# MULTI-TASK REINFORCEMENT LEARNING WITH MIXTURE OF ORTHOGONAL EXPERTS

**Ahmed Hendawy**[1,2*]**, Jan Peters**[1,2,3,4]**, Carlo D'Eramo**[1,2,5]
[1]Department of Computer Science, TU Darmstadt, Germany
[2]Hessian Center for Artificial Intelligence (Hessian.ai), Germany
[3]Center for Cognitive Science, TU Darmstadt, Germany
[4]German Research Center for AI (DFKI), Systems AI for Robot Learning
[5]Center for Artificial Intelligence and Data Science, University of Würzburg, Germany

## ABSTRACT

Multi-Task Reinforcement Learning (MTRL) tackles the long-standing problem of endowing agents with skills that generalize across a variety of problems. To this end, sharing representations plays a fundamental role in capturing both unique and common characteristics of the tasks. Tasks may exhibit similarities in terms of skills, objects, or physical properties while leveraging their representations eases the achievement of a universal policy. Nevertheless, the pursuit of learning a shared set of diverse representations is still an open challenge. In this paper, we introduce a novel approach for representation learning in MTRL that encapsulates common structures among the tasks using orthogonal representations to promote diversity. Our method, named Mixture Of Orthogonal Experts (MOORE), leverages a Gram-Schmidt process to shape a shared subspace of representations generated by a mixture of experts. When task-specific information is provided, MOORE generates relevant representations from this shared subspace. We assess the effectiveness of our approach on two MTRL benchmarks, namely MiniGrid and MetaWorld, showing that MOORE surpasses related baselines and establishes a new state-of-the-art result on MetaWorld.[1]

## 1 INTRODUCTION

Reinforcement Learning (RL) has shown outstanding achievements in a wide array of decision-making problems, including Atari games (Mnih et al., 2013; Hessel et al., 2018a), board games (Silver et al., 2016; 2017), high-dimensional continuous control (Schulman et al., 2015; 2017; Haarnoja et al., 2018), and robot manipulation (Yu et al., 2019). Despite the success of RL, generalizing the learned policy to a broader set of related tasks remains an open challenge. Multi-Task Reinforcement Learning (MTRL) is introduced to scale up the RL framework, holding the promise of enabling learning a universal policy capable of addressing multiple tasks concurrently. To this end, sharing knowledge is vital in MTRL (Teh et al., 2017; D'Eramo et al., 2020; Sodhani et al., 2021; Sun et al., 2022). However, deciding upon the kind of knowledge to share and the set of tasks to share that knowledge is crucial for designing an efficient MTRL algorithm. Human beings exhibit remarkable adaptability across a multitude of tasks by mastering some essential skills as well as having the intuition of physical laws. Similarly, MTRL can benefit from sharing representations that capture unique and diverse properties across multiple tasks, easing the learning of an effective policy.
Recently, sharing compositional knowledge (Devin et al., 2017; Calandriello et al., 2014; Sodhani et al., 2021; Sun et al., 2022) has shown potential as an effective form of knowledge transfer in MTRL. For example, Devin et al. (2017) investigate knowledge transfer challenges between distinct robots and tasks by sharing a modular policy structure. This approach leverages task-specific and robot-specific modules, enabling effective transfer of knowledge. Nevertheless, this approach requires manual intervention to determine the allocation of responsibilities for each module, given some prior knowledge. In contrast, we aim for an end-to-end approach that implicitly learns

---

[*]Ahmed Hendawy (`ahmed.hendawy@tu-darmstadt.de`) is the corresponding author.
[1]The code is available at `https://github.com/AhmedMagdyHendawy/MOORE`.

and shares the prominent components of the tasks for acquiring a universal policy. Furthermore, CARE (Sodhani et al., 2021) adopt a different strategy by focusing on learning representations of different skills and objects encountered by the tasks by utilizing context information. However, there is no inherent guarantee of achieving diversity among the learned representations. In this work, our goal is to *ensure the diversity of the learned representations to maximize the representation capacity and avoid collapsing to similar representations.*

Consequently, we propose a novel approach for representation learning in MTRL to share a set of representations that capture unique and common properties shared by all the tasks. To ensure the richness and diversity of these shared representations, our approach solves a constrained optimization problem that orthogonalizes the representations generated by a mixture of experts via the application of the Gram-Schmidt process, thus favoring dissimilarity between the representations. Hence, we name our approach, **M**ixture **O**f **OR**thogonal **E**xperts (MOORE). Notably, the orthogonal representations act as bases that span a subspace of representations leveraged by all tasks where task-relevant properties can be interpolated. More formally, we show that these orthogonal representations are a set of orthogonal vectors belonging to a particular Riemannian manifold where the inner product is defined, known as Stiefel manifold (James, 1977). Interestingly, the Stiefel manifold has recently drawn substantial attention within the field of machine learning (Ozay & Okatani, 2016; Huang et al., 2018a; Li et al., 2019; Chaudhry et al., 2020). For example, several works focus on enhancing the generalization and stability of neural networks by solving an optimization problem to learn parameters in the Stiefel manifold. Another line of work aims to reduce the redundancy of the learned features by forcing the weights to inhabit the Stiefel manifold. Additionally, Chaudhry et al. (2020) propose a continual learning method that forces each task to learn in a different subspace, thus reducing task interference through orthogonalizing the weights.

In this paper, our objective is to ensure diversity among the shared representations across tasks by imposing a constraint that forces these representations to exist within the Stiefel manifold. Thus, we aim to leverage the extracted representations, in combination with deep RL algorithms, to enhance the generalization capabilities of MTRL policies. In the following, we provide a rigorous mathematical formulation of the MTRL problem, inspired by Sodhani et al. (2021), where latent representations belong to the Stiefel manifold. Then, we devise our MOORE approach for obtaining orthogonal task representations through the application of a Gram-Schmidt process on the latent features extracted from a mixture of experts. We empirically validate MOORE on two widely used and challenging MTRL problems, namely MiniGrid (Chevalier-Boisvert et al., 2023) and Meta-World (Yu et al., 2019), comparing to recent baselines for MTRL. Remarkably, MOORE establishes a new state-of-the-art performance on the MetaWorld MT10 and MT50 collections of tasks.

To recap, the *contribution* of this work is twofold: (i) We propose a mathematical formulation, named Stiefel Contextual Markov Decision Process (SC-MDP), that defines the MTRL problem where the state is encoded in the Stiefel manifold through a mapping function. (ii) We devise a novel representation learning method for Multi-Task Reinforcement Learning that leverages a modular structure of the shared representations to capture common components across multiple tasks. Our approach, named MOORE, learns a mixture of orthogonal experts by encouraging diversity through the orthogonality of their corresponding representations. Our approach outperforms related baselines and achieves state-of-the-art results on the MetaWorld benchmark.

## 2 Preliminaries

A Markov Decision Process (MDP) (Bellman, 1957; Puterman, 1995) is a tuple $\mathcal{M} = < \mathcal{S}, \mathcal{A}, \mathcal{P}, r, \rho, \gamma >$, where $\mathcal{S}$ is the state space, $\mathcal{A}$ is the action space, $\mathcal{P} : \mathcal{S} \times \mathcal{A} \to \mathcal{S}$ is the transition distribution where $\mathcal{P}(s'|s, a)$ is the probability of reaching $s'$ when being in state $s$ and performing action $a$, $r : \mathcal{S} \times \mathcal{A} \to \mathbb{R}$ is the reward function, $\rho$ is the initial state distribution, and $\gamma \in (0, 1]$ is the discount factor. A policy $\pi$ maps each state $s$ to a probability distribution over the action space $\mathcal{A}$. The goal of RL is to learn a policy that maximizes the expected cumulative discounted return $J(\pi) = \mathbb{E}_\pi[\sum_{t=0}^{\infty} \gamma^t r(s_t, a_t)]$. We parameterize the policy $\pi_\theta(a_t|s_t)$ and optimize the parameters $\theta$ to maximize $J(\pi_\theta) = J(\theta)$.

### 2.1 Multi-Task Reinforcement Learning

In MTRL, the agent interacts with different tasks $\tau \in \mathcal{T}$, where each task $\tau$ is a different MDP $\mathcal{M}^\tau = < \mathcal{S}^\tau, \mathcal{A}^\tau, \mathcal{P}^\tau, r^\tau, \rho^\tau, \gamma^\tau >$. The goal of MTRL is to learn a single policy $\pi$ that maximizes

the expected accumulated discounted return averaged across all tasks $J(\theta) = \sum_\tau J_\tau(\theta)$. Tasks can differ in one or more components of the MDP. A class of problems in MTRL assumes only a change in the reward function $r^\tau$. This can be exemplified by a navigation task where the agent learns to reach multiple goal positions or a robotic manipulation task where the object's position changes. In this class, the goal position is usually augmented to the state representation. Besides the reward function, a bigger set of problems deals with changes in other components. In this category, tasks access a subset of the state space $\mathcal{S}^\tau$, while the true state space $\mathcal{S}$ is unknown. For example, learning a universal policy that performs multiple manipulation tasks interacting with different objects (Yu et al., 2019). Task information should be provided either in the form of task ID (e.g., one-hot vector) or metadata, e.g., task description (Sodhani et al., 2021).

Following Sodhani et al. (2021), we define the MTRL problem as a Block Contextual Markov Decision Process (BC-MDP). It is defined by 5-tuple $< \mathcal{C}, \mathcal{S}, \mathcal{A}, \gamma, \mathcal{M}^{'} >$ where $\mathcal{C}$ is the context space, $\mathcal{S}$ is the true state space, $\mathcal{A}$ is the action space, while $\mathcal{M}^{'}$ is a mapping function that provides the task-specific MDP components given the context $c \in \mathcal{C}$, $\mathcal{M}^{'}(c) = \{r^c, \mathcal{P}^c, \mathcal{S}^c, \rho^c\}$. As of now, we refer to the task $\tau$ and its components by the context parameter denoted as $c$.

# 3 RELATED WORKS

Sharing knowledge among tasks is a key benefit of MTRL over single-task learning, as broadly analyzed by several works that propose disparate ways to leverage the relations between tasks (D'Eramo et al., 2020; Sodhani et al., 2021; Sun et al., 2022; Calandriello et al., 2014; Devin et al., 2017; Yang et al., 2020). Among many, D'Eramo et al. (2020) establish a theoretical benefit of MTRL over single-task learning as the number of tasks increases, and Teh et al. (2017) learn individual policies while sharing a prior among tasks. However, naive sharing may exhibit negative transfer since not all knowledge should be shared by all tasks. An interesting line of work investigates the task interference issue in MTRL from the gradient perspective. For example, Yu et al. (2020) propose a gradient projection method where each task's gradient is projected to an orthogonal direction of the others. Nevertheless, these approaches are sensitive to the high variance of the gradients. Another approach, known as PopArt (Hessel et al., 2018b), examines task interference focusing on the instability caused by different reward magnitudes, addressing this issue by a normalizing technique on the output of the value function.

Recently, sharing knowledge in a modular form has been advocated for reducing task interference. Yang et al. (2020) share a base model among tasks while having a routing network that generates task-specific models. Moreover, Devin et al. (2017) divide the responsibilities of the policy by sharing two policies, allocating one to different robots and the other to different tasks. Additionally, Sun et al. (2022) propose a parameter composition technique where a subspace of policy is shared by a group of related tasks. Moreover, CARE Sodhani et al. (2021) highlight the importance of using metadata for learning a mixture of state encoders shared among tasks, based on the claim that the learned encoders produce diverse and interpretable representations through an attention mechanism. Despite the potential of this work, the method is highly dependent on the context information as shown in this recent work (Cheng et al., 2023). However, we argue that all of these approaches lack the guarantee of learning diverse representations.

In this work, we promote diversity across a mixture of experts by enforcing orthogonality among their representations. The mixture-of-experts has been well-studied in the RL literature (Akrour et al., 2021; Ren et al., 2021). Moreover, some works dedicate attention to maximizing the diversity of the learned skills in RL (Eysenbach et al., 2018). Previous works leverage orthogonality for disparate purposes (Mackey et al., 2018). For example, Bansal et al. (2018) promote orthogonality on the weights by a regularized loss to stabilize training in deep convolutional neural networks. Similarly, Huang et al. (2018a) employ orthogonality among the weights for stabilizing the distribution of activation in neural networks. In the context of MTRL, Paredes et al. (2012) enforce the representation obtained from a set of similar tasks to be orthogonal to the one obtained from selected tasks known to be unrelated. Recently, Chaudhry et al. (2020) alleviate catastrophic forgetting in continual learning by organizing task representations in orthogonal subspaces. Finally, Mashhadi et al. (2021) favor diversity in an ensemble of learners via a Gram-Schmidt process. As opposed to it, our primary focus lies in the acquisition of a set of orthogonal representations that span a subspace shared by a group of tasks where task-relevant representations can be interpolated.

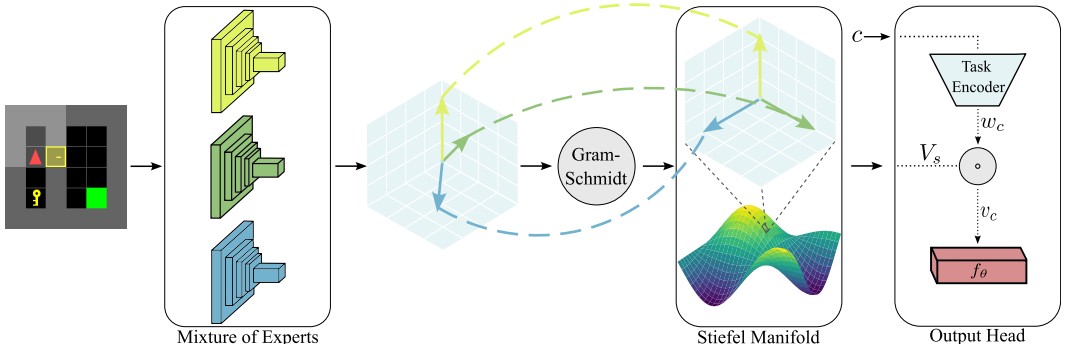

Figure 1: MOORE illustrative diagram. A state $s$ is encoded as a set of representations using a mixture of experts. The Gram-Schmidt process orthogonalizes the representations to encourage diversity. Then, the output head processes the representations $V_s$ by interpolating the task-specific representations $v_c$ using the task-specific weights $w_c$, from which we compute the output using the output function $f_\theta$. In our approach, we employ this architecture for both the actor and the critic.

## 4 SHARING ORTHOGONAL REPRESENTATIONS

We aim to obtain a set of rich and diverse representations that can be leveraged to find a universal policy that accomplishes multiple tasks. To this end, we propose to enforce the orthogonality of the representations extracted by a mixture of experts.

In the following, we first provide a mathematical formulation from which we derive our approach. In particular, we highlight the connection between our method and the Stiefel manifold theory (Huang et al., 2018b; Chaudhry et al., 2020; Li et al., 2020), together with the description of the role played by the Gram-Schmidt process. Then, we proceed to devise our novel method for Multi-Task Reinforcement Learning on orthogonal representation obtained from a mixture of experts.

### 4.1 ORTHOGONALITY IN CONTEXTUAL MARKOV DECISION PROCESSES

We study the optimization of a policy $\pi$, given a set of $k$-orthonormal representations in $\mathbb{R}^d$ for the state $s$. We define the orthonormal representations of state $s$ as a matrix $V_s = [v_1, ..., v_k] \in \mathbb{R}^{d \times k}$ where $v_i \in \mathbb{R}^d, \forall i \leq k$. It can be shown that the orthonormal representations $V_s$ belong to a topological space known as the Stiefel manifold, a smooth and differentiable manifold largely used in machine learning (Huang et al., 2018b; Chaudhry et al., 2020; Li et al., 2020).

**Definition 4.1** *(Stiefel Manifold) Stiefel manifold $\mathcal{V}_k(\mathbb{R}^d)$ is defined as the set of all orthonormal $k$-vectors in the Euclidean space $\mathbb{R}^d$, where $k \leq d$, $\mathcal{V}_k(\mathbb{R}^d) = \{V_s \in \mathbb{R}^{d \times k} : V_s^T V_s = I_k, \forall s \in \mathcal{S}\}$.*

Under this lens, our goal can be interpreted as finding a set of orthogonal representations belonging to the Stiefel manifold that capture the common characteristics in the true state space $\mathcal{S}$. Thus, we propose a novel MDP formulation for MTRL, which we call a Stiefel Contextual Markov Decision Process (SC-MDP), that is inspired by the BC-MDP introduced in Sodhani et al. (2021). An SC-MDP includes a function that maps the state $s$ to $k$-orthonormal representations $V_s \in \mathcal{V}_k(\mathbb{R}^d)$.

**Definition 4.2** *(Stiefel Contextual Markov Decision Process) A Stiefel Contextual Markov Decision Process (SC-MDP) is defined as a tuple $< \mathcal{C}, \mathcal{S}, \mathcal{A}, \gamma, \mathcal{M}', \varphi >$ where $\mathcal{C}$ is the context space, $\mathcal{S}$ is the true state space, $\mathcal{A}$ is the action space. $\mathcal{M}'$ is a function that maps a context $c \in \mathcal{C}$ to MDP parameters and observation space $\mathcal{M}'(c) = \{r^c, \mathcal{P}^c, \mathcal{S}^c, \rho^c\}$, $\varphi$ is a function that maps every state $s \in \mathcal{S}$ to a $k$-orthonormal representations $V_s \in \mathcal{V}_k(\mathbb{R}^d)$, $V_s = \varphi(s)$.*

We define our MTRL policy as $\pi(a|s, c) = f_\theta(\varphi(s) \cdot w_c)$, where $w_c \in \mathbb{R}^k$ is the task-specific weight that combines the $k$-orthogonal representations into a task-relevant one and $f_\theta : \mathbb{R}^d \to \mathbb{R}^{|\mathcal{A}|}$ is an output function with learnable parameters $\theta$ that generates actions from task-specific representations. To leverage a diverse set of representations across tasks, the mapping function $\varphi$

plays a fundamental role. Hence, we approximate $\varphi$ by a mixture of experts $\mathbf{h}_\phi = [h_{\phi_1}, ..., h_{\phi_k}]$ with learnable parameters $\phi = [\phi_1, ..., \phi_k]$ that generate $k$-representations $U_s \in \mathbb{R}^{d \times k}$ for state $s$, while ensuring that the generated representations are orthogonal to enforce diversity. Conveniently, this objective finds a rigorous formulation as a constrained optimization problem where we impose a hard constraint to enforce orthogonality:

$$\max_{\Theta = \{\phi, \theta\}} \quad J(\Theta)$$
$$\text{s.t.} \quad \mathbf{h}_\phi^T(s) \, \mathbf{h}_\phi(s) = I_k \quad \forall s \in \mathcal{S}, \tag{1}$$

where $I_k \in \mathbb{R}^{k \times k}$ is the identity matrix. Instead of solving the constrained optimization problem in Eq. 1, we ensure the diversity across experts through the application of the Gram-Schmidt (GS) process to orthogonalize the $k$-representations $U_s$.

**Definition 4.3** *(Gram-Schmidt Process) Gram-Schmidt process is a method for orthogonalizing a set of linearly independent $\mathcal{U} = \{u_1, ..., u_k : u_i \in \mathbb{R}^d, \forall i \leq k\}$. It maps the vectors to a set of $k$-orthonormal vectors $\mathcal{V} = \{v_1, ..., v_k : v_i \in \mathbb{R}^d, \forall i \leq k\}$ defined as*

$$v_k = u_k - \sum_{i=1}^{k-1} \frac{\langle v_i, u_k \rangle}{\langle v_i, v_i \rangle} v_i. \tag{2}$$

where the representation of the $i$-th expert $u_i$ is projected in the orthogonal direction to the subspace spanned by the representations of all $i-1$ experts. Therefore, we apply the GS process to map the generated representations by the mixture of experts $U_s = \mathbf{h}_\phi(s)$ to a set of orthonormal representations $V_s = GS(U_s)$, satisfying the hard constraint in Eq. 1.

## 4.2 Multi-Task Reinforcement Learning with Orthogonal Representations

Following our policy $\pi(a|s, c)$, each task can interpolate its relevant representation from the subspace spanned by the $k$-orthonormal representations $V_s$. We train a task encoder to produce the task-specific weights $w_c \in \mathbb{R}^k$ given task information (e.g. task ID). The orthonormal representations are combined using the task-specific weight to produce relevant representations $v_c \in \mathbb{R}^d$ to the task as $v_c = V_s \cdot w_c$. The interpolated representation $v_c$ captures the relevant components of the task that can be utilized by the RL algorithm and fed to an output function $f_\theta$. The output function can be learned for each task separately (multi-head) or shared by all tasks (single-head) to compute the action components given the representations $v_c$. Similarly, the same policy (actor) structure (Alg. 1) can be used for the critic (Alg. 2). In conclusion, this approach results in a **M**ixture **O**f **OR**thogonal **E**xperts, thus, we call it MOORE, whose extracted representation is used to learn a universal policy for MTRL. A visual demonstration of our approach is shown in Fig.1.
We adopt two different RL algorithms, namely Proximal Policy Optimization (PPO) and Soft Actor-Critic (SAC), with the purpose of demonstrating that our approach is agnostic to the used RL algorithms. PPO (Schulman et al., 2017) is a policy gradient algorithm that has the merit of obtaining satisfactory performance in a wide range of problems while being easy to implement. It is a first-order method that enhances the policy update given the current data by limiting the deviation of the new policy from the current one. Moreover, we integrate our approach to SAC, a high-performing off-policy RL algorithm that leverages entropy maximization to enhance exploration.

## 5 Experimental Results

In this section, we evaluate MOORE against related baselines on two challenging MTRL benchmarks, namely MiniGrid (Chevalier-Boisvert et al., 2023), a set of visual goal-oriented tasks, and MetaWorld (Yu et al., 2019), a collection of robotic manipulation tasks. The objective is to assess the adaptability of our approach in handling different types of state observations and tackling a variable number of tasks. Moreover, the flexibility of MOORE is evinced by using it for on-policy (PPO for MiniGrid) and off-policy RL (SAC for MetaWorld) algorithms. Additionally, we conduct ablation studies that support the effectiveness of MOORE in various aspects. We assess the following points: the benefit of using Gram-Schmidt to impose diversity across experts, the quality of the learned representations, as well as the transfer capabilities, and the interpretability of the diverse experts.

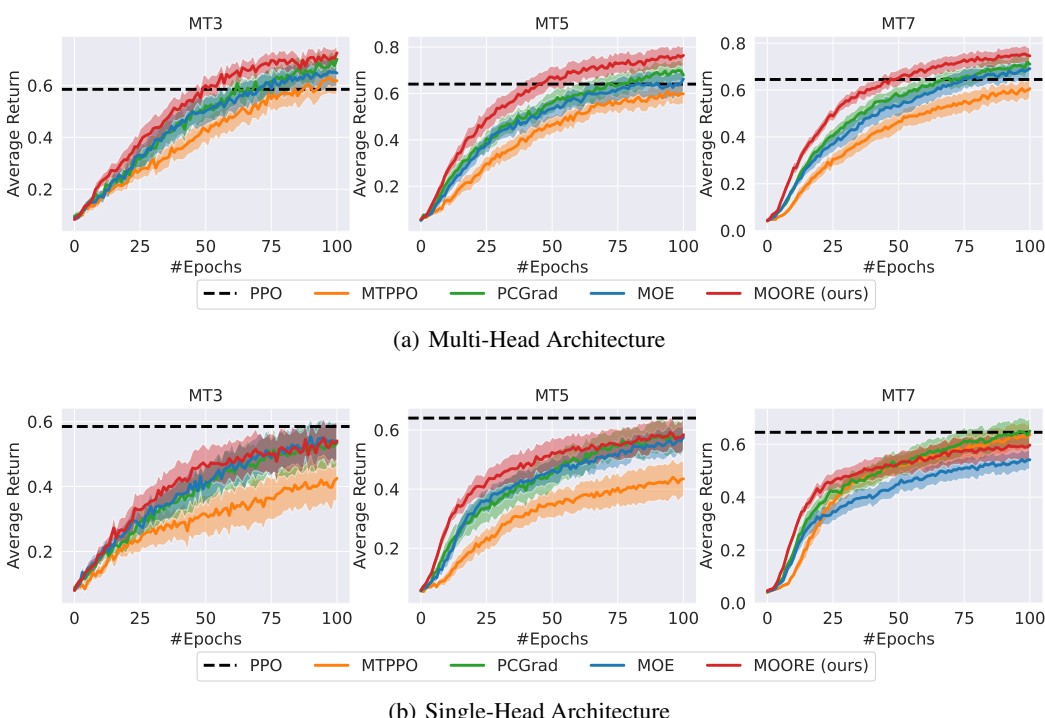

(a) Multi-Head Architecture

(b) Single-Head Architecture

Figure 2: Average return on the three MTRL scenarios of MiniGrid. We utilize both multi-head and single-head architectures for our approach MOORE as well as the related baselines. For MOORE, MOE and PCGrad, the number of experts $k$ is 2, 3, and 4 for MT3, MT5, and MT7, respectively. The black dashed line represents the final single-task performance of PPO averaged across all tasks. For the evaluation metric, we compute the accumulated return averaged across all tasks. We report the mean and the 95% confidence interval across 30 different runs.

## 5.1 MINIGRID

We consider different tasks in MiniGrid (Chevalier-Boisvert et al., 2023), a suite of 2D goal-oriented environments that requires solving different mazes while interacting with objects like doors, keys, or boxes of several colors, shapes, and roles. MiniGrid offers a visual representation of the state, which we adopt for our multi-task setting. We consider the multi-task setting from Jin et al. (2023) that includes three multi-task scenarios. The first scenario, **MT3**, involves the three tasks: LavaGap, RedBlueDoors, and Memory; the second scenario, **MT5**, includes the five tasks: DoorKey, Lava-Gap, Memory, SimpleCrossing, and MultiRoom. Finally, **MT7** comprises the seven tasks: DoorKey, DistShift, RedBlueDoors, LavaGap, Memory, SimpleCrossing, and MultiRoom. In Sec. A.1, we provide descriptions and more details for the tasks.

We compare MOORE against four baselines. The first one is **PPO**, considered a reference for comparing to single-task performance. The second baseline is **Multi-Task PPO (MTPPO)**, an adaptation of PPO (Schulman et al., 2017) for MTRL. Then, we consider **MOE**, which employs a mixture of experts to generate representations without enforcing diversity across experts. Additionally, we have **PCGrad** (Yu et al., 2020), which is an MTRL approach that tackles the task interference issue by manipulating the gradients. We integrate PCGrad on top of the MOE baseline for a fair comparison. As for the MTRL architecture, we utilize multi-head and single-head architectures for all methods, showing their average return across all tasks in Fig. 2(a), and Fig. 2(b) respectively. MOORE outperforms the aforementioned baselines in almost all the MTRL scenarios. Notably, our method exhibits faster convergence than the baselines. It is interesting to observe that MOORE outperforms the single-task performance with a significant margin in comparison to the other baselines (Fig.2(a)), which is usually considered as an upper-bound of the MTRL performance in previous works. This highlights the quality of the learned representations and the role of MOORE representation learning process in MTRL.

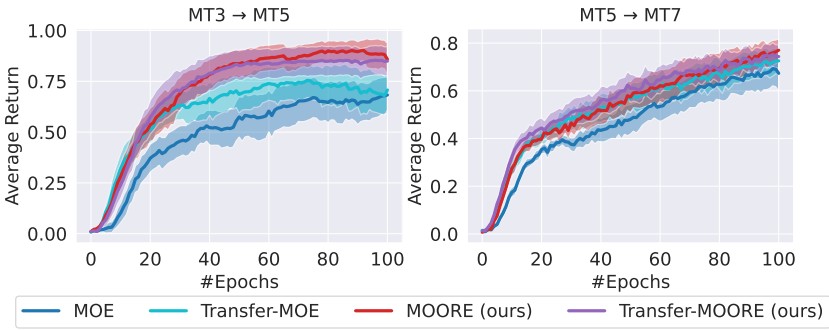

Figure 3: Evaluating MOORE against MOE on the transfer setting. The study is conducted on the two transfer learning scenarios in MiniGrid, employing a multi-head architecture. The number of experts $k$ is 2 and 3 for MT3 → MT5 and MT5 → MT7, respectively. For the evaluation metric, we compute the accumulated return averaged across all tasks. We report the mean and the 95% confidence interval across 30 different runs.

### 5.1.1 ABLATION STUDIES

**Transfer Learning.** We examine the advantage of transferring the trained experts on a set of *base* tasks to *novel* tasks in order to assess the quality and generalization of these learned experts in comparison to the MOE baseline. We refer to the transfer variant of our approach as **Transfer-MOORE** while **Transfer-MOE** for the baseline. Moreover, we include the performance of MOORE and MOE as a MTRL reference for learning the *novel* tasks from scratch, completely isolated from the *base* tasks. In Fig. 3, we show the empirical results on two transfer learning scenarios where we transfer a set of experts learned on MT3 to MT5 (**MT3 → MT5**) and on MT5 to MT7 (**MT5 → MT7**). MT3 is a subset of MT5, while MT5 is a subset of MT7. First, we train on the *base* tasks, and then we transfer the learned experts (frozen) to

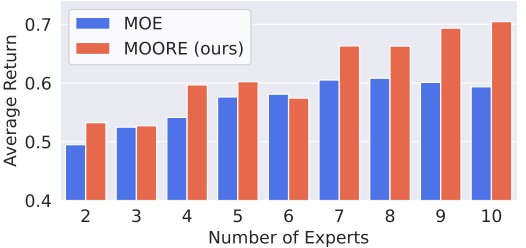

Figure 4: Ablation study on the effect of changing the number of experts. We compare the performance of MOE and MOORE (ours) on MiniGrid MT7 using a single-head architecture. We report the mean of the evaluation metric across 30 seeds. For the evaluation metric, we compute the accumulated return averaged across all tasks.

the *novel* tasks (the difference between the two sets). As illustrated in Fig. 3, Transfer-MOORE outperforms Transfer-MOE in the two scenarios, showing the quality of the learned representations in the context of transfer learning. Moreover, the study demonstrates the ability of our approach as an effective MTRL algorithm that provides competitive results against the transfer variant (Transfer-MOORE). In contrast, MOE struggles to beat the transfer variant as in the MT3 → MT5 scenario. Consequently, this study advocates the diversification of the shared representations in transfer learning and MTRL. We highlight more details in B.2.

**Number of Experts.** Additionally, we focus on the impact of changing the number of experts on the performance of our approach, as well as on MOE. In Fig. 4, we consider different numbers of experts on the MT7 scenario. We observe the effect of utilizing more experts in MOORE algorithm compared to MOE. The study shows that MOORE exhibits a noticeable advantage, on average, for an increasing number of experts. On the contrary, a slower enhancement of the performance is encountered by MOE. It is also worth noting that the performance of MOORE with $k = 4$ slightly outperforms MOE with $k = 10$ while being comparable to MOE with $k = 8$ (MOE best setting). This supports our claim about efficiently utilizing expert capacity through enforcing diversity.

## 5.2 METAWORLD

Finally, we evaluate our approach on another challenging MTRL setting with a large number of manipulation tasks. We benchmark against MetaWorld (Yu et al., 2019), a widely adopted robotic manipulation benchmark for Multi-Task and Meta Reinforcement Learning. We consider the **MT10**

| Total Env Steps | 1M | 2M | 3M | 5M | 10M | 15M | 20M |
|---|---|---|---|---|---|---|---|
| SAC (Yu et al., 2019) | 10.0±8.2 | 17.7±2.1 | 18.7±1.1 | 20.0±2.0 | 48.0±9.5 | 57.7±3.1 | 61.9±3.3 |
| MTSAC (Yu et al., 2019) | 34.9±12.9 | 49.3±9.0 | 57.1±9.8 | 60.2±9.6 | 61.6±6.7 | 65.6±10.4 | 62.9±8.0 |
| SAC + FiLM (Perez et al., 2017) | 32.7±6.5 | 46.9±9.4 | 52.9±6.4 | 57.2±4.2 | 59.7±4.6 | 61.7±5.4 | 58.3±4.3 |
| PCGrad (Yu et al., 2020) | 32.2±6.8 | 46.6±9.3 | 54.0±8.4 | 60.2±9.7 | 62.6±11.0 | 62.6±10.5 | 61.7±10.9 |
| Soft-Module (Yang et al., 2020) | 24.2±4.8 | 41.0±2.9 | 47.4±5.3 | 51.4±6.8 | 53.6±4.9 | 56.6±4.8 | 63.0±4.2 |
| CARE (Sodhani et al., 2021) | 26.0±9.1 | 52.6±9.3 | 63.8±7.9 | 66.5±8.3 | 69.8±5.1 | 72.2±7.1 | 76.0±6.9 |
| PaCo (Sun et al., 2022) | 30.5±9.5 | 49.8±8.2 | 65.7±4.5 | 64.7±4.2 | 71.0±5.5 | 81.0±5.9 | 85.4±4.5 |
| MOORE (ours) | **37.2±9.9** | **63.0±7.2** | **68.6±6.9** | **77.3±9.6** | **82.7±7.3** | **88.2±5.6** | **88.7±5.6** |

Table 1: Results on MetaWorld MT10 (Yu et al., 2019) with random goals (MT10-rand). The results of the baselines are from Sun et al. (2022). MOORE uses $k = 4$ experts. For all methods, we report the mean and standard deviation of the evaluation metric across 10 different runs. The evaluation metric is the average success rate across all tasks. We highlight with bold text the **best** result.

and **MT50** settings, where a single robot has to perform 10 and 50 tasks, respectively.

For the baselines, we compare our approach against the following algorithms. First, **SAC** (Haarnoja et al., 2018) is the off-policy RL algorithm that is trained on each task separately, thus being a reference for the single-task setting. Second, **Multi-Task SAC (MTSAC)** is the adaptation of SAC to the MTRL setting, where we employ a single-head architecture with a one-hot vector concatenated with the state. Then, **SAC+FiLM** is a task-conditional policy that employs the FiLM module (Perez et al., 2017). Furthermore, **PCGrad** (Yu et al., 2020) is an MTRL approach that tackles the task interference issue by manipulating the gradients. **Soft-Module** (Yang et al., 2020) utilizes a routing network that proposes weights for soft combining of activations for each task. **CARE** (Sodhani et al., 2021) is an attention-based approach that learns a mixture of experts for encoding the state while utilizing context information. Finally, **PaCo** (Sun et al., 2022) is the state-of-the-art method for MetaWorld that learns a compositional policy where task-specific weights are utilized for interpolating task-specific policies. Our approach uses a similar framework as in the MiniGrid experiment and employs a multi-head architecture.

Following Sun et al. (2022), we benchmark against variants of the MT10 and MT50 scenarios, MT10-rand and MT50-rand, where each task is trained with random goal positions. The goal position is concatenated with the state representation. As a performance metric, we compute the success rate averaged across all tasks. We run our approach for 10 different runs and report their mean and standard deviations of the metric, similar in Sun et al. (2022). As stated in Tab. 1, MOORE outperforms all the baselines regarding sample efficiency and asymp-

| Algorithms | Success Rate (20M) |
|---|---|
| MTSAC (Yu et al., 2019) | 49.3±1.5 |
| SAC + FiLM (Perez et al., 2017) | 36.5±12.0 |
| CARE (Sodhani et al., 2021) | 50.8±1.0 |
| PaCo (Sun et al., 2022) | 57.3±1.3 |
| MOORE (ours) | **72.9±3.3** |

Table 2: Results on MetaWorld MT50 (Yu et al., 2019) with random goals (MT50-rand). The results of the baselines are from Sun et al. (2022). MOORE uses $k = 6$ experts.

totic performance. Moreover, in Tab. 2, our approach shows significant final performance, indicating the scalability of MOORE to a large number of tasks. It is important to mention that all baselines use tricks to enhance the stability of the learning process. For instance, PaCo avoids task and gradient explosion by proposing two empirical tricks, named *loss maskout* and *w-reset*, where *pruning* every task loss that reaches above a certain threshold, besides *resetting* the task-specific weight for that task. Also, as in Sun et al. (2022), the other baselines resort to more expensive tricks, such as terminating and re-launching the training session when a loss explosion is encountered. On the contrary, our approach does not need such tricks to improve the stability of the learning process, which can indicate the stability of the chosen architecture and the importance of learning distinct experts.

### 5.2.1 ABLATION STUDIES

**Diversity.** Similarly, we want to evince the advantage of favoring diversity across experts. We evaluate MOORE against MOE, a baseline that uses the same architecture of MOORE but without the Gram-Schmidt process. We evaluate MOORE against MOE on the two MTRL scenarios of MetaWorld, MT10-rand and MT50-rand. In Fig. 5(a), MOORE exhibits superior sample-efficiency compared to MOE. Moreover, MOORE significantly outperforms the baseline also in MT50-rand

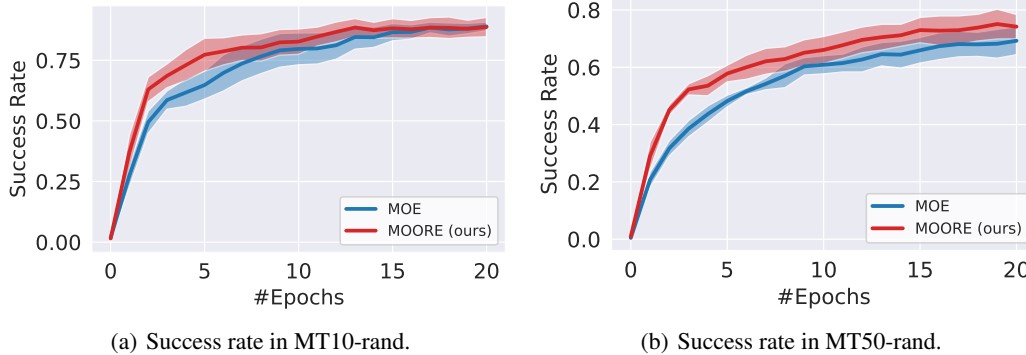

(a) Success rate in MT10-rand.  (b) Success rate in MT50-rand.

Figure 5: (a) Success rate on MetaWorld MT10-rand comparing MOORE, against MOE, using 4 experts. (b) Success rate on MetaWorld MT50-rand comparing MOORE, against MOE, given 6 experts. We show the average success rate across all tasks and the $95\%$ confidence interval across 10 and 5 different runs for MT10-rand and MT50-rand, respectively.

(Fig. 5(b)), evincing the scalability of our approach to large-scale MTRL problems. This study illustrates the importance of enforcing diversity across experts in MTRL algorithms.

**Interpretability.** Additionally, we verify the interpretability of the learned representations. Fig. 6 shows an application of PCA on the learned task-specific weights $w_c$ that interpolate the representations of the experts. On the one hand, the *pick-place* task is close to the *peg-insert-side* since both tasks require picking up an object. On the other hand, the weights of *door-open* and *window-open* tasks are similar as they share the open skill. Therefore, enforcing diversity across experts distributes the responsibilities across them in capturing common components across tasks (e.g., objects or skills). This confirms that the learned experts have some roles that can be interpretable.

## 6 CONCLUSION AND DISCUSSION

We proposed a novel MTRL approach for diversifying a mixture of shared experts across tasks. Mathematically, we formulate our objective as a constrained optimization problem where a hard constraint is explicitly imposed to ensure orthogonality between the representations. As a result, the orthogonal representations live on a smooth and differentiable manifold called the Stiefel manifold. We formulate our MTRL as a novel contextual MDP while mapping each state to the Stiefel manifold using a mapping function, which we learn through a mixture of experts while enforcing orthogonality across their representations with the Gram-Schmidt process, hence satisfying the hard constraint. Our approach demonstrates superior performance against related baselines on two challenging MTRL baselines.

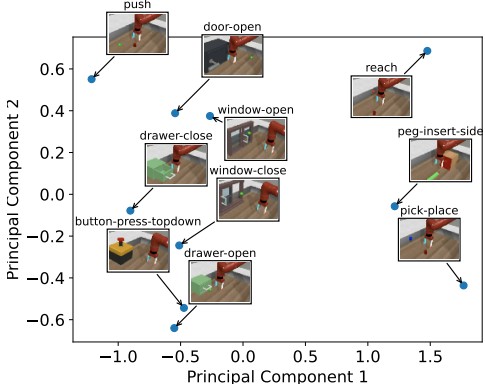

Figure 6: Principle Component Analysis (PCA) on the task-specific weights learned by MOORE on MetaWorld MT10-rand for a run with 100% success rate across all tasks.

Taking advantage of all the experts during inference, our approach has the limitation of potentially suffering from high time complexity compared to a sparse selection of few experts. This leads to a trade-off between the representation capacity and time complexity, which could be investigated in the future by a selection of a few orthogonal experts. In addition to our transfer learning study, we are interested in investigating extensions of our approach into a continual learning setting.

ACKNOWLEDGMENTS

We want to thank Aliaa Khalifa for her support in writing the paper and Firas Al-Hafez for his feedback on the method. This work was funded by the German Federal Ministry of Education and Research (BMBF) (Project: 01IS22078). This work was also funded by Hessian.ai through the project 'The Third Wave of Artificial Intelligence – 3AI' by the Ministry for Science and Arts of the state of Hessen. Calculations for this research were conducted on the Lichtenberg high-performance computer of the TU Darmstadt and the Intelligent Autonomous Systems (IAS) cluster at TU Darmstadt.

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

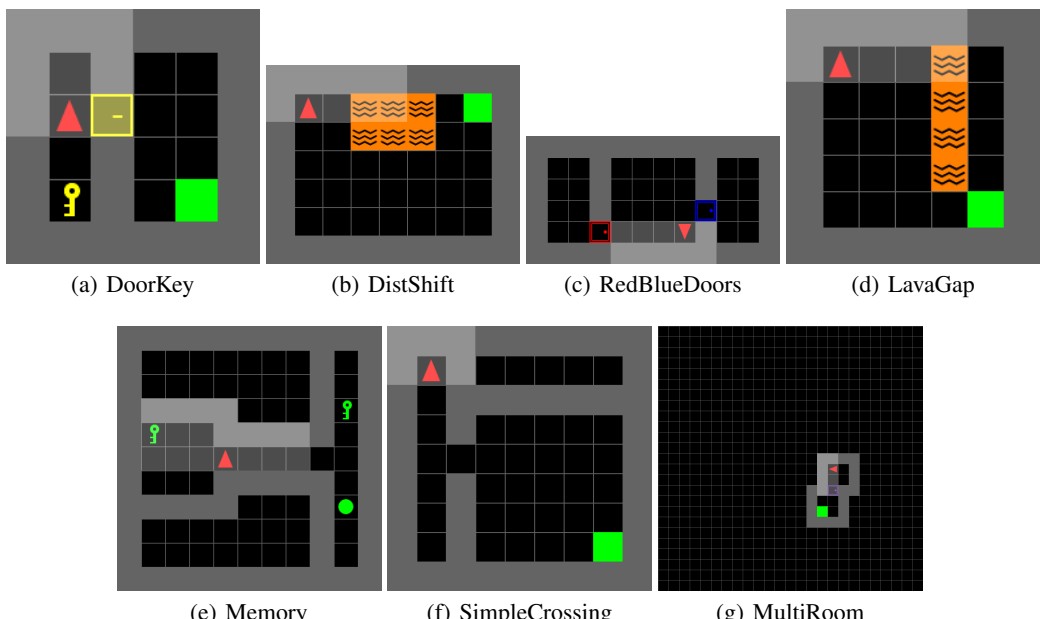

(a) DoorKey     (b) DistShift     (c) RedBlueDoors     (d) LavaGap

(e) Memory     (f) SimpleCrossing     (g) MultiRoom

Figure 7: MiniGrid (Chevalier-Boisvert et al., 2023) Tasks, where the red triangle represents the agent, and the green square refers to the goal.

# A  ADDITIONAL DETAILS ON THE EXPERIMENTS

In this section, we elaborate on the implementation details of our approach, MOORE, for benchmarking against MiniGrid (Chevalier-Boisvert et al., 2023) and MetaWorld (Yu et al., 2019). Besides, we provide additional ablation studies that demonstrate various aspects of our approach. In this work, we used Mushroom-RL (D'Eramo et al., 2021) as the RL library.

## A.1  MINIGRID

### A.1.1  ENVIRONMENT DETAILS

MiniGrid (Chevalier-Boisvert et al., 2023) is a collection of 2D goal-oriented environments where the agent learns how to solve different mazes while interacting with various objects in terms of shape, color, and role. The library of MiniGrid provides multiple choice for state representation. For our MTRL setting, we adopt the visual representation of the state where a 3-dimensional input of shape 7x7x3 is provided. As mentioned in Sec. 5.1, our MTRL setting consists of three scenarios that include seven tasks in total that are distributed differently. A render example of each task is demonstrated in Fig. 7. Additionally, the description of each task is provided in Tab. 3.

| Task | Description |
|------|-------------|
| DoorKey | Use the key to open the door and then get to the goal. |
| DistShift | Get to the green goal square. |
| RedBlueDoors | Open the red door and then the blue door |
| LavaGap | Avoid the lava and get to the green goal square. |
| Memory | Go to the matching object at the end of the hallway |
| SimpleCrossing | Find the opening and get to the green goal square |
| MultiRoom | Traverse the rooms to get to the goal. |

Table 3: MiniGrid (Chevalier-Boisvert et al., 2023) task descriptions.

A.1.2 IMPLEMENTATION DETAILS

**RL algorithm.** We use PPO (Schulman et al., 2017), which is considered a state-of-the-art on-policy RL algorithm on many benchmarks. Moreover, it has been used in the official paper of the MiniGrid benchmark (Chevalier-Boisvert et al., 2023). We adapt PPO to the MTRL setting by computing the loss functions of both the actor and critic averaged on transitions sampled from all tasks. We refer to this adapted algorithm as MTPPO. In Tab. 4, we highlight the important hyperparameters needed to reproduce the results on MiniGrid.

| Hyperparameter | Value |
|---|---|
| General Hyperparameters | |
| Discount factor $\gamma$ | 0.99 |
| Number of environments | [3,5,7] |
| Steps per environment | 1 step per 1 environment |
| Number of epochs | 100 |
| Steps per epoch | 2000 |
| Train frequency | 2000 |
| Number of episodes for evaluation | 16 |
| PPO Hyperparameters | |
| Lambda coefficient in GAE formula | 0.95 |
| Entropy term coefficient | 0.01 |
| Clipping Epsilon | 0.2 |
| Number of epochs for Policy | 8 |
| Batch Size for Policy | 256 |
| Number of epochs for Critic | 1 |
| Batch Size for Critic | 2000 |
| Critic Loss | Mean Squared Error |
| Optimizer | Adam |
| Learning rate for Policy | 0.001 |
| Learning rate for Critic | 0.001 |

Table 4: MiniGrid (Chevalier-Boisvert et al., 2023) hyperparameters.

**Architecture.** The network architecture consists of two main parts, a *representation block*, and an *output head*. The *representation block* is agnostic to the context $c$. The role of the *representation block* is to encode the state $s$. On the other hand, the *output head* includes an *output function* for generating the network output. In general, we use a similar network architecture for the actor and the critic.

For single-expert approaches (PPO and MTPPO), the *representation block* consists of a single Convolutional Neural Network (CNN) to encode the visual representation of the state to a latent space. For multiple-experts approaches (MOORE, MOE, and PCGrad), $k$-CNNs are used to represent the mixture of experts responsible for encoding the state as $k$-representations in the *representation block*.

For MTRL approaches, the *output function* can utilize a single-head $f_\theta$ or a multi-head $f_\theta = [f_{\theta_1}, .., f_{\theta_{|\mathcal{C}|}}]$ architecture. For the single-head architecture, we condition the network on the context by concatenating the context $c$ (one-hot vector) to the output of the *representation block*. On the other hand, for the multi-head architecture, we select a task-specific *output function* $f_{\theta_c}$ given the context $c$.

For multiple-experts approaches, in addition to the *output function*, the *output head* includes a *task-encoder*. Given a context $c$, the *task-encoder* generates a task-specific weight $w_c$ responsible for combining the output of the *representation block* $V_s$ to produce the task-specific representation $v_c$.

In Tab. 5, we illustrate the hyperparameters of both the *representation block* and the *output head*. It is worth noting that MOORE, MOE, and PCGrad linearly combine the generated representations from different experts before applying the last activation function of the representation block $v_c = \text{Tanh}(V_s \cdot w_c)$. Moreover, the whole architecture is trained end-to-end, including the *task-encoder*.

| Hyperparameter | Value |
|---|---|
| Representation Block | |
| Number of Experts ($k$) | {MT3: $k = 2$, MT5: $k = 3$, MT7: $k = 4$} |
| Number of convolution layers | 3 |
| Channels per layer | [16, 32, 64] |
| Kernel size | [(2,2), (2,2), (2,2)] |
| Activation functions | [ReLU, ReLU, Tanh] |
| Output Function | |
| Number of linear layers | 2 (x number of tasks $|\mathcal{T}|$) |
| Number of output units | [128, $|\mathcal{A}|$ for actor and 1 for critic] |
| Activation functions | [Tanh, Linear] |
| Task Encoder | |
| Number of linear layers | 1 |
| Number of output units | Number of Experts ($k$) |
| Use bias | False |
| Activation function | Linear |

Table 5: Actor and Critic Architecture for PPO

---

**Algorithm 1** MOORE for Actor

**Require:** Mixture of experts $\mathbf{h}_\phi$, state $s$, context $c$, task-specific weights $w_c$, output function $f_\theta$.
1: $U_s = \mathbf{h}_\phi(s)$
2: $V_s = GS(U_s)$       ▷ Apply Eq. 2
3: $v_c = V_s \cdot w_c$
4: $a \sim f_\theta(v_c)$
5: **Return:** a

**Algorithm 2** MOORE for Critic

**Require:** Mixture of experts $\mathbf{h}_\phi$, state-action $(s, a)$, context $c$, task-specific weights $w_c$, output function $f_\theta$.
1: $U_{s,a} = \mathbf{h}_\phi(s, a)$
2: $V_{s,a} = GS(U_{s,a})$       ▷ Apply Eq. 2
3: $v_c = V_{s,a} \cdot w_c$
4: $q = f_\theta(v_c)$
5: **Return:** q

## A.2 METAWORLD

### A.2.1 ENVIRONMENT DETAILS

MetaWorld (Yu et al., 2019) is a suite of many robotic manipulation tasks. All tasks require dealing with one or two objects. Moreover, they are similar in terms of the state space's dimensionality, yet the state components' semantics differ. The state space consists of the following: the 3D position of the end effector, a normalized measure of how much the gripper is open, the 3D position of the first object, the quaternion of the first object (4D), as well as the 3D position and quaternion of the second object (zeroed out, if not needed). Two consecutive data frames are stacked together, in addition to the 3D goal position, forming a 39-dimensional state space. On the other hand, the action space is the same, representing the 3D change of the end effector in addition to the normalized torque applied by the gripper. We benchmark our approach against the MT10 and MT50 scenarios. Following Sun et al. (2022), we randomize the goal or object positions across all tasks and refer to them as MT10-rand and MT50-rand.

### A.2.2 IMPLEMENTATION DETAILS

**RL algorithm.** In this benchmark, we use SAC (Haarnoja et al., 2018), a state-of-the-art off-policy algorithm that enhances the exploration of the agent by maximizing the entropy. Similar to Yu et al. (2019); Sun et al. (2022), we adapt SAC by computing the actor and the critic losses averaged on transitions sampled from all tasks. We have a replay buffer for each task from which we sample transitions equally. In addition, we disentangle the temperature parameter of SAC by learning separate temperature parameters for each task. We refer to this adapted algorithm as MTSAC. In Tab. 6, we list the hyperparameters required for reproducing our results on MetaWorld.

**Architecture.** Similar to MiniGrid, we use a network architecture that consists of a *representation block* and an *output head*. We made a couple of changes for MetaWorld. For instance, the actor and the critic slightly differ since the action is concatenated with the state for computing the Q values

in the critic. As a result, the *representation block* is responsible for encoding the state-action space. Another difference is that we use a Dense Neural Network (DNN) to represent the *representation block*. Consequently, we use k-DNNs to represent the mixture of experts for MOORE and MOE. Finally, we adopted a multi-head architecture for the *output function* where we use the context $c$ to select the corresponding task-specific output function $f_{\theta_c}$.

It is worth mentioning that the results of the baselines in Tab. 1 and Tab. 2 are borrowed from Sun et al. (2022). The implementation details of the baselines can be found in Yu et al. (2019); Sun et al. (2022). We demonstrate the MOORE algorithm for the actor and the critic in Alg. 1 and Alg. 2, respectively. Similarly, MOE follows the same procedure but without the Gram-Schmidt process in line 2.

| Hyperparameter | Value |
|---|---|
| General Hyperparameters | |
| Horizon | 150 |
| Discount factor $\gamma$ | 0.99 |
| Number of environments | 10 |
| Steps per environment | 1 step per 1 environment |
| Number of epochs | 20 |
| Steps per epoch | 100000 |
| Train frequency | 1 |
| Number of episodes for evaluation | 10 |
| SAC Hyperparameters | |
| Batch Size | 128 |
| Critic Loss | Mean Squared Error |
| Disentangled temperature Alpha $\alpha$ | True |
| Optimizer | Adam |
| Learning rate for Policy | $3 \times 10^{-4}$ |
| Learning rate for Critic | $3 \times 10^{-4}$ |
| Learning rate for Alpha | $1 \times 10^{-4}$ |
| Policy minimum standard | $e^{-10}$ |
| Policy maximum standard | $e^2$ |
| Soft target interpolation | $5 \times 10^{-3}$ |
| Exploration steps | 1500 |
| Replay buffer steps | $1 \times 10^6$ |

Table 6: MetaWorld (Yu et al., 2019) Hyperparameters.

| Hyperparameter | Value |
|---|---|
| Representation Block | |
| Number of Experts ($k$) | {MT10: $k = 4$, MT50: $k = 6$} |
| Number of Linear layers | 3 |
| Number of output units | [400, 400, 400] |
| Activation functions | [ReLU, ReLU, Linear] |
| Output Block | |
| Number of linear layers | 1 (x number of tasks $|\mathcal{T}|$) |
| Number of output units | [$|\mathcal{A}|$ for actor and 1 for critic] |
| Activation functions | Linear |
| Task Encoder | |
| Number of linear layers | 1 |
| Number of output units | Number of Experts ($k$) |
| Use bias | False |
| Activation function | Linear |

Table 7: Actor and Critic Architecture for SAC

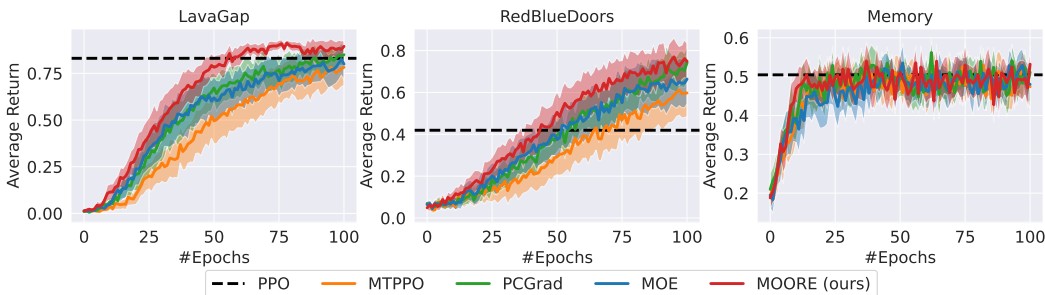

Figure 8: Individual task average return on the MT3 scenario of MiniGrid. We utilize the multi-head architecture for our approach MOORE as well as the related baselines. For MOORE, MOE, and PCGrad, the number of experts $k$ is 2. The black dashed line represents the final single-task performance of PPO averaged across all tasks. For the evaluation metric, we compute the accumulated return averaged across all tasks. We report the mean and the 95% confidence interval across 30 different runs.

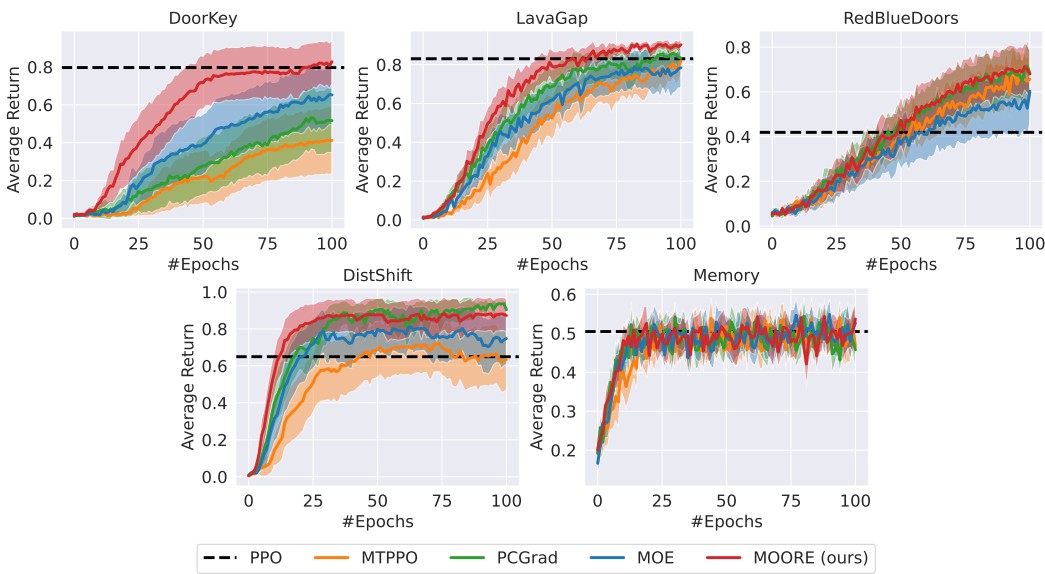

Figure 9: Individual task average return on the MT5 scenario of MiniGrid. We utilize the multi-head architecture for our approach MOORE as well as the related baselines. For MOORE, MOE, and PCGrad, the number of experts $k$ is 3. The black dashed line represents the final single-task performance of PPO averaged across all tasks. For the evaluation metric, we compute the accumulated return averaged across all tasks. We report the mean and the 95% confidence interval across 30 different runs.

## B  ADDITIONAL EMPIRICAL RESULTS

### B.1  MINIGRID

In Sec. 5.1, we present the performance averaged across all the tasks. Here, we want to show the individual task performance of all three scenarios of MiniGrid.

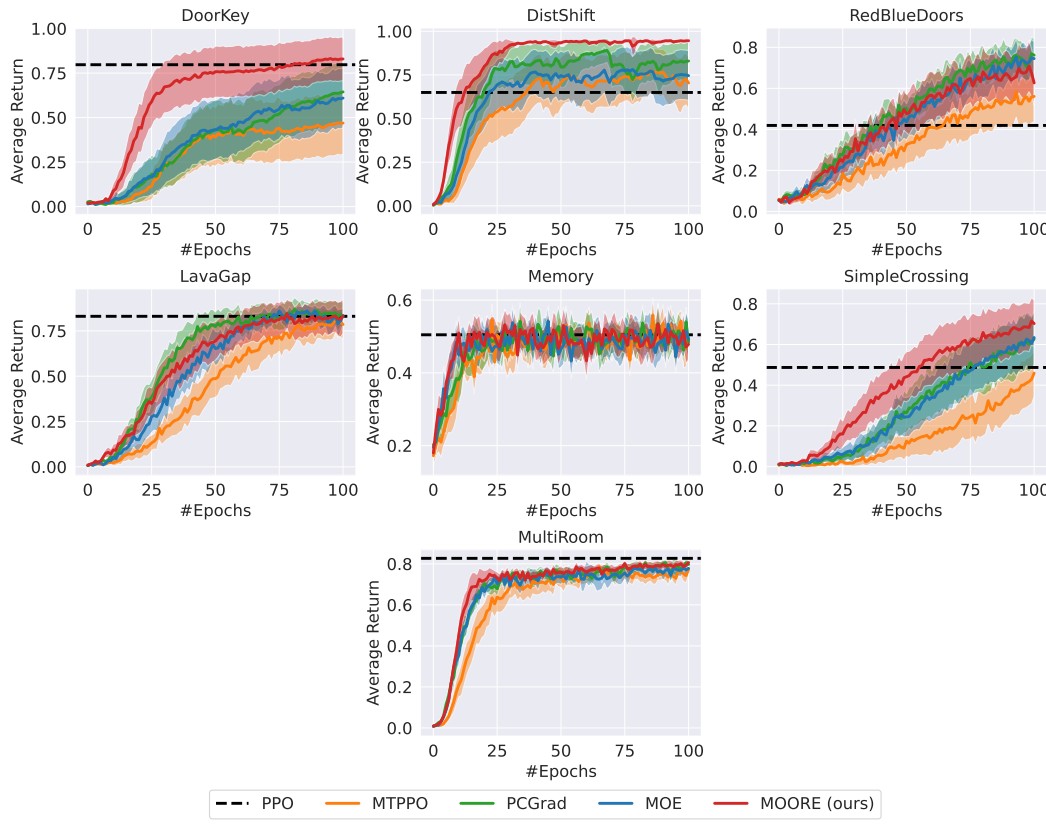

Figure 10: Individual task average return on the MT7 scenario of MiniGrid. We utilize the multi-head architecture for our approach MOORE as well as the related baselines. For MOORE, MOE, and PCGrad, the number of experts $k$ is 4. The black dashed line represents the final single-task performance of PPO averaged across all tasks. For the evaluation metric, we compute the accumulated return averaged across all tasks. We report the mean and the 95% confidence interval across 30 different runs.

## B.2 TRANSFER LEARNING WITH MOORE

Furthermore, we discuss the experimental details of the Transfer Learning ablation study in Fig. 3. In this study, we assess the transfer capability of our approach in utilizing the diverse representations learned on a set of *base* tasks for a set of *novel* but related tasks. We evaluate our approach, MOORE, against the MOE baseline on MiniGrid. We refer to the transfer learning adaptation of our approach as **Transfer-MOORE** and **Transfer-MOE** for the MOE baseline.

We conducted two experiments based on the sets of tasks defined on MiniGrid (MT3, MT5, and MT7). In Fig. 3, we show the empirical results on two transfer learning scenarios where we transfer a set of experts learned on MT3 to MT5 (**MT3 → MT5**) and on MT5 to MT7 (**MT5 → MT7**). It is worth noting that MT3 is a subset of MT5, and MT5 is a subset of MT7. The base tasks are the MT3 and MT5 for MT3 → MT5 and MT5 → MT7, respectively, while the novel tasks are the difference between the corresponding sets. For instance, in the MT3→MT5 scenario, the base tasks are LavaGap, RedBlueDoors, and Memory (common for MT3 and MT5), while having DoorKey, and MultiRoom as novel tasks (only in MT5).

For Transfer-MOORE, we train on the base tasks; then, we use the learned mixture of experts in a frozen state to learn the novel ones. On the contrary, MOORE is only trained on novel tasks from scratch. This also holds for MOE and Transfer-MOE. In this study, we employ a multi-head architecture for the actor and critic. Hence, each task has a decoupled output head from other tasks, easing the transfer learning experiment. However, they all share the representation stage (mixture

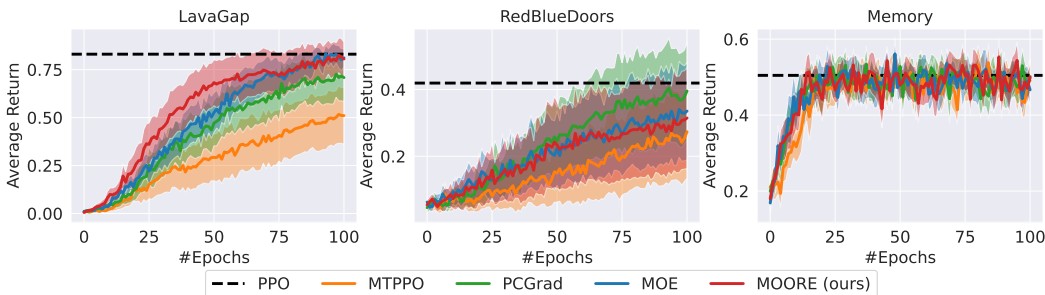

Figure 11: Individual task average return on the MT3 scenario of MiniGrid. We utilize the single-head architecture for our approach MOORE as well as the related baselines. For MOORE, MOE, and PCGrad, the number of experts $k$ is 2. The black dashed line represents the final single-task performance of PPO averaged across all tasks. For the evaluation metric, we compute the accumulated return averaged across all tasks. We report the mean and the 95% confidence interval across 30 different runs.

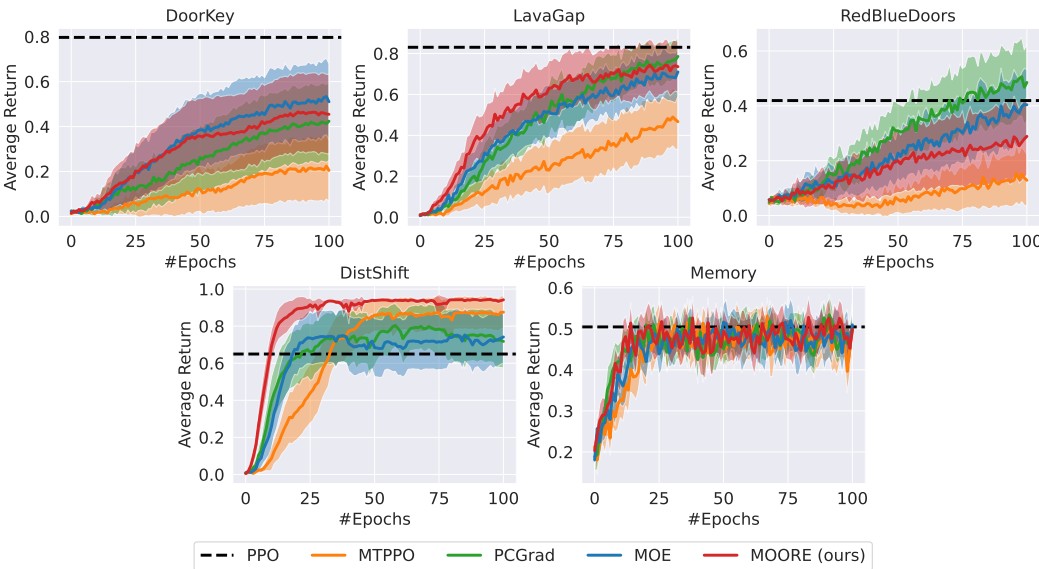

Figure 12: Individual task average return on the MT5 scenario of MiniGrid. We utilize the single-head architecture for our approach MOORE as well as the related baselines. For MOORE, MOE, and PCGrad, the number of experts $k$ is 3. The black dashed line represents the final single-task performance of PPO averaged across all tasks. For the evaluation metric, we compute the accumulated return averaged across all tasks. We report the mean and the 95% confidence interval across 30 different runs.

of experts). We add randomly initialized output heads to learn the novel tasks while keeping the mixture of experts frozen. For **MT3 → MT5**, the number of experts $k$ is 2. On the other hand, for **MT5 → MT7**, we use 3 experts.

## B.3 COSINE SIMILARITY

We investigate the ability of MOORE to diversify the shared representations, compared to relaxing the hard constraint in Eq. 1. Therefore, we replace the hard constraint with a regularization term

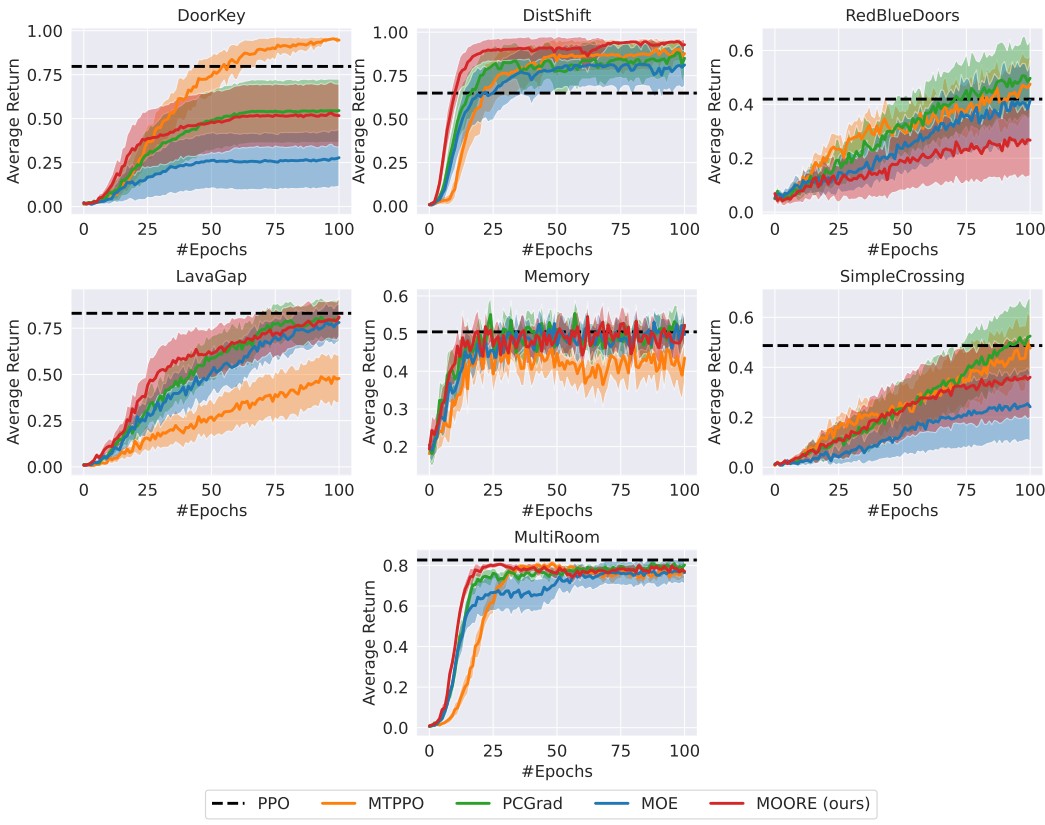

Figure 13: Individual task average return on the MT7 scenario of MiniGrid. We utilize the single-head architecture for our approach MOORE as well as the related baselines. For MOORE, MOE, and PCGrad, the number of experts $k$ is 4. The black dashed line represents the final single-task performance of PPO averaged across all tasks. We show the accumulated return averaged across all tasks. We report the mean and the 95% confidence interval across 30 different runs.

equivalent to a cosine similarity loss computed over the set of representations:

$$l_{\text{reg}} = \mathbb{E}_{s \in \mathcal{S}} \left[ \mathbf{h}_\phi(s)^T \mathbf{h}_\phi(s) - I_k \right]. \tag{3}$$

The regularization loss is optimized jointly with the primary objective, where we weigh the contribution of this regularization loss by 1. We benchmark MOORE against the **Cosine-Similarity** on the three scenarios of MiniGrid. As shown in Fig. 14, MOORE outperforms the baseline across all settings, highlighting the advantage of using Gram-Schmidt in diversifying the experts over regularization-based techniques. In addition, our approach is *hyperparameter-free*, contrary to the regularization-based techniques that require delicate hyperparameter tuning to not interfere with the main loss function, which is usually the case.

## B.4 INFLUENCE OF THE SINGLE-HEAD ARCHITECTURE ON MOORE

In this section, we discuss the reason behind the degradation in the performance of MOORE when employing a single-head architecture, especially on MT7 (Fig. 2(b)). We argue that the reason is the task interference caused by the single-head architecture since all tasks share the same output function $f_\theta$. MOORE is highly affected by the later output stage, causing a drop in the performance relative to the experiments done with the multi-head architecture. It is worth noting that as the number of tasks increases, the possibility of having task interference increases. This is why the issue is prominent in the MT7 scenario.

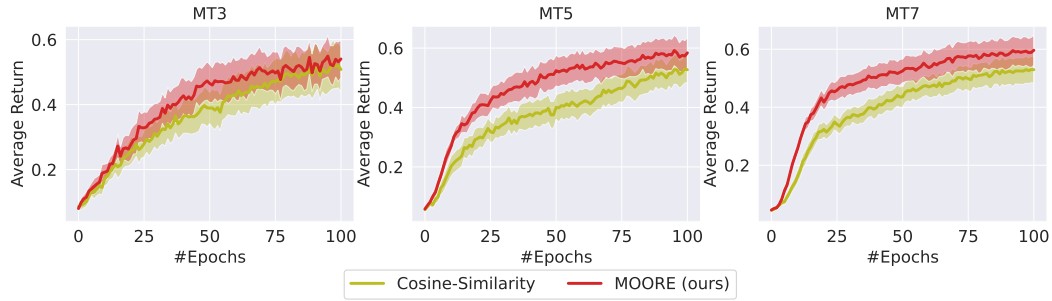

Figure 14: Evaluating the diversity capabilities of our approach, MOORE, against using Cosine-Similarity. The study is conducted on the three MTRL scenarios of MiniGrid employing a single-head architecture. The number of experts $k$ is 2, 3, and 4 for MT3, MT5, and MT7, respectively. For the evaluation metric, we compute the accumulated return averaged across all tasks. We report the mean and the 95% confidence interval across 30 different runs.

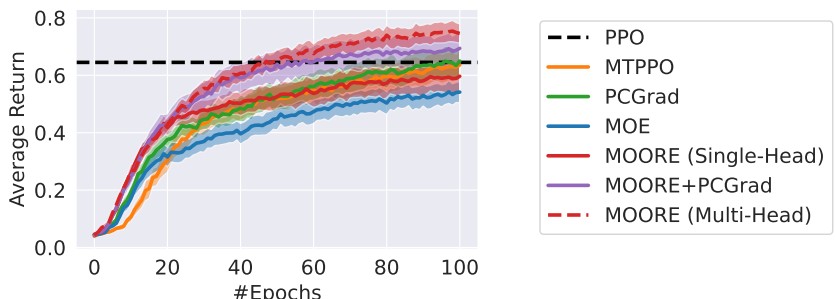

Figure 15: Ablation study on the effect of combining MOORE with PCGrad to reduce the task interference issue in the output stage. All methods employ a single-head architecture except for MOORE (Multi-Head). The study is conducted on the MT7 scenario on MiniGrid. The number of experts $k$ is 4. For the evaluation metric, we compute the accumulated return averaged across all tasks. We report the mean and the 95% confidence interval across 30 different runs.

We have two reasons to support our claim:

- When using a multi-head architecture, MOORE outperforms all the baselines on all of the 3 MiniGrid scenarios. Employing the multi-head architecture decouples the output functions for all tasks, completely removing the task interference in the output stage.

- In Fig. 15, we conduct an ablation study highlighting the effect of combining PCGrad (explicit MTRL method to tackle task interference) and our approach. Since MOORE is orthogonal to PCGrad, we can integrate them easily. This study shows that MOORE+PCGrad outperforms MOORE, PCGrad, MOE, and MTPPO. However, MOORE with multi-head architecture still outperforms MOORE+PCGrad, showing that PCGrad can only partially reduce the interference in the output stage, while MOORE with multi-head architecture removes the interference completely.

## C  COMPUTATION AND MEMORY REQUIREMENTS

The difference between MOORE and MOE is in the Gram-Schmidt stage, where we orthogonalize the $k$ representations. The time complexity of the Gram-Schmidt process is $T = O(k^2 \times d)$ (Golub & Van Loan, 2013; Mashhadi et al., 2021), where $d$ is the representation dimension and $k$ is the number of experts. Our approach MOORE and the baseline MOE belong to the family of soft mixtures of experts since they compute all $k$ representations from all the experts during inference.

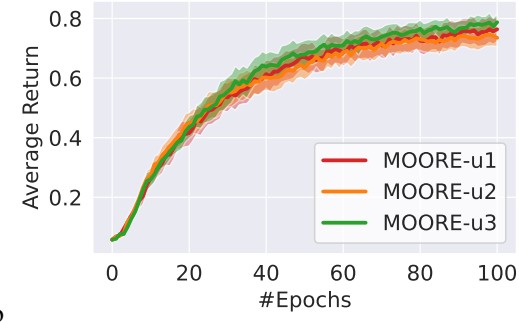

b

Figure 16: Ablation study on the effect of the initial expert selected for the Gram-Schmidt process. In this study, we employ a multi-head architecture. The number of experts $k$ is 3. u1, u2, and u3 are the representations of the three experts before applying the Gram-Schmidt process. For the evaluation metric, we compute the accumulated return averaged across all tasks. We report the mean and the $95\%$ confidence interval across 30 different runs.

On the other hand, one can only select top-k experts based on some weights computed using a gating network as in the direction of sparse mixtures of experts. The trade-off between the representation capacity and time complexity is well-known. As a future work, we can investigate the adaptation of MOORE to pick only a few orthogonal experts, hence lowering the time complexity. MOORE is similar to the MOE baseline regarding the memory required for storing all the experts. It is worth noting that we use fewer experts than PaCo (Sun et al., 2022) in MetaWorld, hence lower memory requirements.

## D  THE GRAM-SCHMIDT PROCESS AND THE INITIAL EXPERT

In MOORE, we consider the first expert's representation as the initial vector for the Gram-Schmidt process. In a normal setting, we can expect the process to yield a different set of orthonormal vectors depending on the initial selected vector. It does not matter in our case since the representations are actually generated from a mixture of experts which are being learned. We conduct an ablation study on the MT5 scenario of MiniGrid, where we utilize 3 experts. We provide variations of MOORE based on the initial vector selected for the Gram-Schmidt process. For instance, MOORE-u1 selects the representation of the first expert u1 as the initial vector of the Gram-Schmidt process (adopted). On the other hand, MOORE-u2 and MOORE-u3 choose the representation of the second u2 and third u3 expert, respectively, as the initial vector for the Gram-Schmidt process. As expected, Fig. 16 shows that the performance is almost identical for different selected initial vectors.

