# OpenReview forum: "Multi-Task Reinforcement Learning with Mixture of Orthogonal Experts"
_ICLR.cc/2024/Conference — ICLR 2024 poster_

### Official Review · Reviewer_duFe · 2023-10-28

**Soundness:** 3 good
**Presentation:** 2 fair
**Contribution:** 2 fair
**Rating:** 6
**Confidence:** 3

**Summary:**

This paper studies learning a good representation for multi-task setting. Their definition of a good representation acroos tasks crucially relies on the idea of capturing unique and common properties shared by all the tasks. To do this they solve a constrained optimization problem that orthogonalizes the representations generated by a mixture of experts by applying the Gram-Schmidt orthogonalization technique. This results in ensuring independence between the representations. They draw a nice connection between their orthogonal representations to the Stiefel manifold, a particular type of Riemannian manifold where the inner product is defined. In fact they define a Steifel Contextual MDP (SC-MDP) in definition 4.2 that maps any arbitrary state $s$ to a vector representation in Steifel Manifold (consequently these vector representations are orthonormal). They learn this mapping function $\phi$ (called experts) from the data and while doing so concurrently enforce a constraint so that the learned representations are orthonormal (see eq(1)). Finally, to enforce diversity across experts, they apply the Gram-Schmidt process to orthogonalize the k-representations. They generate the final RL policy by combining these representations with a linear parametrization. Finally, they conduct several empirical evaluation of their algorithm in several standard RL benchmarks.

**Strengths:**

1) Learning a diverse, meaningful, and common representation across several tasks is an important area of research.
2) Their approach is somewhat novel where they enforce as they draw this nice connection between the Stiefel manifold and orthonormal representations (in the Stiefel manifold) leads to diverse and meaningful representations across tasks.
3) Their approach is simple and leads to good empirical performance in some benchmarks.
4) The Mixture of Experts (MOE) is their main competitor which employs a mixture of experts without enforcing diversity. Principally they show that as you increase the number of experts the average return of MOE is less than MOORE. They also show how task-specific weights align with representations.

**Weaknesses:**

1) While they draw some connection to the Stiefel manifold, they do not analyze theoretically any properties of their representations that might give some meaningful insight as to why these orthonormal representations are useful.
2) The paper is not clear about the computation cost of the constrained optimization problem in eq(1).
3) The paper lacks some more experimental discussion to prove conclusively that the orthonormal representations are helping over their closest competitor MOE.

**Questions:**

1) There is no discussion in the paper regarding the computational cost of the constrained optimization in (1). How hard is it to solve and enforce this constraint?
2) I understand that your approach enhances the diversity of feature representation which in turn leads to a good exploration of the state space. How do you ensure the balance between exploration and exploitation? Do you rely on the PPO and SAC to do a standard exploration/exploitation with the diverse learnt features?
3) The discussion/implementation details between single and multi-head version of MOORE is missing. This seems important because in MT7 (figure 2) MOORE is outperformed by PC-grad and MTPPO? Why?
4) It was vaguely referred to in section 5.1 (below figure 4) that MOORE has a faster convergence rate than other baselines. Can you elaborate on this? Do you believe that diverse representations (the key thrust of this paper) lead to a faster convergence?
5) The ablation study is somewhat confusing. Observe that the ablation study is done on single-head architecture and you already show in MT7 (figure 1) that MOE is outperformed by MOORE. However, PC-grad and MTPPO outperform MOORE in figure 1 (single head). So it is not clear whether even PC-grad and MTPPO are learning a diverse set of representations. Can you elaborate on this?

**Details Of Ethics Concerns:**

Not applicable.

---

> ### Author Response · Authors · 2023-11-16
>
> > Computation cost of enforcing the constraint
>
> Instead of solving the expensive constrained optimization problem at each time step, we take into account the hard constraint by applying the Gram-Schmidt process on the generated representations from the experts to orthogonalize them, hence satisfying the constraint at each time step. The time complexity of the Gram-Schmidt process is T = $O(k^{2}\times d)$, where $d$ is the representation dimension and $k$ is the number of experts.
>
> > Exploration and Exploitation
>
> MOORE does not influence the exploration-exploitation trade-off. Instead, the main focus is extracting diverse representations of common objects and skills shared across the tasks. We rely on the RL algorithm (SAC, PPO) to handle the exploration aspect.
>
> > Single-head and Multi-head
>
> The discussion of single-head and multi-head is highlighted in the appendix (A.1.2). In the revised version, we enhanced the discussion in the Appendix and added a clear sentence in section 4.2 of the methodology. Here, we will provide a clear distinction between the single-head and multi-head architectures. In multi-task, there are two common architectures: single-head, and multi-head. The difference between them is highly dependent on how we condition our network. In single-head architecture, the context $c$ (e.g. task-id) is usually concatenated with the input and fed to the network. Since MOORE has a task-agnostic representation block, we concatenate the context $c$ with the output of the representation block (aggregated representation) $v_{c}$. On the contrary, the multi-head consists of multiple task-specific output modules $f_{\theta} = [f_{\theta_{1}}, .., f_{\theta_{|\mathcal{C}|}}]$ where the context $c$ is responsible for selecting the corresponding task-specific output module $f_{\theta_{c}}$. This is also valid for the baselines.
>
> > Figure 2 discussion
>
> Given the multiple evidence of the effectiveness of MOORE in MiniGrid, MetaWorld MT10, and the new results in MetaWorld MT50, we believe we have demonstrated the superiority of our approach compared to baselines on large-scale MTRL problems. We believe that the single case where MOORE performs worse is negligible.
>
> > Fast convergence
>
> In MTRL, sharing representations is key to learning multiple tasks using the same resources. Learning representations and skills shared across tasks can help in solving difficult tasks. We believe the MOE baseline eventually learns independent experts at the end of the training, yet it requires more samples to do so. It needs to focus on learning diverse experts and avoid any representation collapse while enhancing the quality of representation for every expert. In MOORE, we force diversity through the application of the Gram-Schmidt process, hence the network will only focus on learning high-quality representations for each expert. We argue this is the reason why our approach exhibits high sample efficiency in comparison to the other baselines.
>
> > Stiefel manifold and theoretical analysis
>
> Stiefel manifold has been well-studied in machine learning [1,2]. Our goal in proposing the SC-MDP formulation is to connect the RL literature to the machine learning one that investigate the importance of the Stiefel manifold. For instance, [2] use the Stiefel manifold to alleviate the catastrophic forgetting in the continual learning scenario. We believe that our study could be of inspiration for future investigations on continual learning.
>
> > More discussion about the results and the importance of the orthogonality
>
> Our experiments show that MOORE consistently outperforms MOE on the challenging MTRL tasks of MiniGrid and MetaWorld MT10. Nevertheless, in the revised version of the paper, we now provide new results on the very challenging MetaWorld MT50, where MOORE again outperforms the MOE, even by a large margin. We believe that this can convince the Reviewer about the superiority of MOORE w.r.t. MOE and about the importance of diversity in the representation.
>
> > Single-head and ablation studies
>
> In Fig. 4, we do the ablation study on MT7 since we can evaluate MOORE on a large number of experts before reaching the case where we have an expert for each task ($k=7$). Besides, we select the single-head architecture due to time reasons, since the experiments are faster than the multi-head ones. As the Reviewer points out, MOORE performs slightly worse than the baseline in this setting. However, we consider this lower performance to be negligible, especially considering the superiority of MOORE in all the other settings.
>
> [1] Huang, Lei, et al. "Orthogonal weight normalization: Solution to optimization over multiple dependent stiefel manifolds in deep neural networks." AAAI (2018).
>
> [2] Chaudhry, Arslan, et al. "Continual learning in low-rank orthogonal subspaces." NeurIPS (2020).

---

> > ### Comment · Reviewer_duFe · 2023-11-21
> > **Answer to Rebuttal**
> >
> > Thank you for your rebuttal and answering some of my questions.
> > - Thank you for clarifying the computational complexity. One of my concerns from your section 5.1.1 is that as you increase the number of experts you get a better performance against MOE (fig 5) at the cost of the computational complexity. My guess is that this is also dependent on the task at hand. How do you choose the right number of experts?
> > - Thank you for clarifying the exploration-exploitation question.
> > - I understand the difference between single-head vs multi-head architecture. My main concern is that in MT7 (figure 2) MOORE is outperformed by PC-grad and MTPPO? Why? Is this something to do with PPO?
> > - Thanks for conducting the additional experiment on MT50.

---

> > > ### Author Response · Authors · 2023-11-21
> > >
> > > > Thank you for clarifying the computational complexity. One of my concerns from your section 5.1.1 is that as you increase the number of experts you get a better performance against MOE (fig 5) at the cost of the computational complexity. My guess is that this is also dependent on the task at hand. How do you choose the right number of experts?
> > >
> > > Yes, there is a trade-off between the time complexity and the performance as a factor of the number of experts $k$. The number of experts $k$ is a hyperparameter that needs to be tuned to balance this trade-off. The time complexity affects the training and inference time. In MTRL, at each training step, we sample a transition from all tasks. Hence, the time complexity of the single training step is also a function of the number of tasks. Another important factor is the memory requirements. Since multi-task learning is about using the resources of the model efficiently by sharing, the multi-task model should be smaller than the single-task model where each task has its own model (representation and output modules) without sharing.
> > >
> > > Consequently, we select the number of experts based on the aforementioned factors. Across all experiments, we tried to choose a number of experts which is below the number of tasks but still gives a reasonably good performance. For instance, we selected $2$ experts (minimum possible number) for MT3 ($3$ tasks), $3$ experts for MT5 ($5$ tasks), and $4$ experts for MT7 ($7$ tasks). In MiniGrid, the memory requirement is the main factor for selection. On the other hand, in MetaWorld, we selected $4$ experts for MT10-rand and $6$ experts for MT50-rand. We could not increase it too much, especially on MT50-rand, due to the computation complexity of the sampling from a large number of tasks. Then, the selection is based on the memory requirements as well as the computation complexity of the training process.
> > >
> > > We could have selected more experts as shown in Fig. 4 (Fig.5 in the original submission) to get better performance. We believe that as the number of experts increases, the more diverse representations of common properties between tasks that can be shared.
> > >
> > > > I understand the difference between single-head vs multi-head architecture. My main concern is that in MT7 (figure 2) MOORE is outperformed by PC-grad and MTPPO? Why? Is this something to do with PPO?
> > >
> > > Thank you for raising this point. We argue that the reason for the performance of MOORE in MT7 with a single-head architecture is the task interference caused by the single-head architecture. After conducting further experiments, we found that, since all tasks share the output module, MOORE is affected by the interference in the later output stage causing the degradation in the performance in compared with the other baselines. It is worth noting that as the number of tasks increases, the possibility of having task interference increases. This is why we can see the issue in the MT7 scenario.
> > >
> > > There are two reasons that support our claim.
> > >
> > > - MOORE outperforms all the baselines on all of the $3$ scenarios when using multi-head architecture. Using a multi-head architecture, removes completely the task interference at the output module since it decouples the output modules for all tasks.
> > > - Second, we have conducted an ablation study on the effect of combining PCGrad (explicit MTRL method to tackle task interference) and our approach. Since MOORE is orthogonal to PCGrad, we can integrate them easily. This study shows that MOORE+PCGrad outperforms MOORE, PCGrad, MOE, and MTPPO, showing comparable performance to MOORE with multi-head architecture. Nevertheless, MOORE with multi-head still outperforms MOORE+PCGrad with single-head since, while PCGrad can only partially reduce the interference, MOORE with multi-head architecture removes the interference completely.
> > >
> > > We have included the experimental results for this analysis in Appendix B.4.
> > >
> > > *We thank the Reviewer for motivating us to include this analysis which we believe helped to improve our paper. In light of these improvements, we kindly ask the Reviewer to consider improving his/her marginally positive assessment of our work.*

---

### Official Review · Reviewer_78VR · 2023-10-31

**Soundness:** 2 fair
**Presentation:** 3 good
**Contribution:** 3 good
**Rating:** 6
**Confidence:** 3

**Summary:**

The paper proposes a novel approach for representation learning in Multi-Task Reinforcement Learning (MTRL) known as MOORE (Mixture Of Orthogonal Experts). The core idea is to share representations across tasks in an orthogonal manner to promote diversity and prevent representations from collapsing. MOORE utilizes the Gram-Schmidt process to shape a shared subspace of representations derived from a mixture of experts. This approach ensures diversity by pushing these representations to exist within the Stiefel manifold, a Riemannian manifold known for its orthogonality properties. The authors validate their method on two MTRL benchmarks: MiniGrid and MetaWorld, and report state-of-the-art performance on the latter.

**Strengths:**

The idea of orthogonalizing shared representations is novel and aligns with the goal of maximizing representation capacity in MTRL.

Leveraging the properties of the Stiefel manifold for enforcing orthogonality is innovative and could have implications beyond MTRL.

The authors have demonstrated strong empirical results, notably achieving state-of-the-art performance on the MetaWorld MT10-rand tasks.

**Weaknesses:**

The author used MoE and Transfer-MoE as its baseline. As far as I know, there are other MoE baselines, like PMOE [1], why the author didn't compare them in experiments.

In environments like DoorKey, as the author showed in Fig. 13, MOORE is worse than some other baselines, why and how this happened needs more discussion.

[1] Ren, Jie, et al. "Probabilistic mixture-of-experts for efficient deep reinforcement learning." arXiv preprint arXiv:2104.09122 (2021).

**Questions:**

See Weakness

---

> ### Author Response · Authors · 2023-11-16
>
> > Other MOE baselines
>
> Thank you for the reference. In this work, we focused on MTRL algorithms like CARE [1] and PaCo which are the state-of-the-art MTRL baselines. Also, they can be considered as MOE baselines since CARE [1] uses a mixture of state encoders (experts) as we do, and PaCo [2] learn a mixture of policies or critics. Our MOE baseline utilizes the same architecture as our approach except for the Gram-Schmidt process stage; thus, we can assess the importance of enforcing diversity across the experts.
>
> > Fig. 13, MOORE is worse than some other baselines, why and how this happened needs more discussion.
>
> Given the multiple evidence of the effectiveness of MOORE in MiniGrid, MetaWorld MT10, and the new results in MetaWorld MT50, we believe we have demonstrated the superiority of our approach compared to baselines on large-scale MTRL problems. We believe that the single case where MOORE performs worse is negligible.
>
> [1] Sodhani, Shagun, et al. "Multi-task reinforcement learning with context-based representations." ICML (2021).
>
> [2] Sun, Lingfeng, et al. "PaCo: Parameter-Compositional Multi-Task Reinforcement Learning." NeurIPS (2022).

---

### Official Review · Reviewer_stvx · 2023-10-31

**Soundness:** 3 good
**Presentation:** 2 fair
**Contribution:** 3 good
**Rating:** 6
**Confidence:** 3

**Summary:**

The authors propose MOORE, a method for learning a common state representation using a mixture of orthogonal experts to be combined with task-specific weights for Multi-Task Reinforcement Learning. Building upon a Block Contextual MDP, a Stiefel-Contextual MDP is defined that formally allows for a compositional policy, taking advantage from a latent state representation retrieved from a mixture of experts. In various empirical studies, this representation is shown to improve the performance of PPO in various discrete tasks, and the performance of SAC in various continuous control tasks. Furthermore, ensuring orthogonality of experts via the Gram-Schmidt Process yields even improved results.

**Strengths:**

The paper tackles a relevant problem and provides a well-motivated solution approach. Overall, the paper is well structured and the presented considerations are theoretically sound. A wide variety of tasks, domains,  action-, and state-spaces, are used for evaluation, using a sufficient amout of baselines and sensible ablations.

**Weaknesses:**

Overall, the clarity of the paper should be improved. Concretely:

- The introduction does not state whether the proposed method is used for learning in a multi-task scenario or to curate diverse tasks for learning. Moreover, providing a motivation why a common state representation across tasks is needed could be useful. Also, the empirical studies could be included with the contributions listed in the introduciton.
- In Section 4, in addition to the visualization of the mixture of experts in Fig. 1, a visualizing the parts of the compositional policy $\pi(a|s)=(V_sw_c)^T\theta$ an their connection would be helpful. To further elaborate on the intertwinement of the components and the proposped process, showcasing an exemplary algorithm might be useful (Section 4.2).
- Regarding the experiments: I assume MOORE to be used in conjuction with PPO for the MiniGrid results (5.1) and with SAC for the MetaWorld results (5.2). Yet this should be clearly stated in the paper. Furthermore, if not introduced in related work, used baselines should be described (e.g., MTPPO, MTSAC, PCGrad, ...). To improve the overall readability, the Figure placement could be improved.
- Limitations should be stated

Related work should be extended with regards to work on mixtures of experts for reinforcmeent learning.
Also, the baselines used later on should be briefly introduced and compared. Further related work regarding diversity-based RL (e.g., B. Eysenbach, A. Gupta, J. Ibarz, and S. Levine, ‘Diversity is All You Need: Learning Skills without a Reward Function’, in International Conference on Learning Representations, 2019.) could also be helpful.

Finally, to foster reproducibility, the authors should consider open-sourcing their implementations upon publication.

**Questions:**

From my understanding, orthogonality is induced via the Gram-Schmidt Process. Could you elaborate on the necessity of the additional hard constraint introduced in Eq. 1?

Regarding the experimental results, why is the performacne of PPO only reported as a horizontal line?

How many experts are used for the Meta World evaluations? How does the number of experts impact the scalabiltiy of the proposed approach? Further elaborations on the computational overhead introduces, as well as possible limitations might be more insightful, than the presented remarks regarding interpretability.

Minor Comments:
- p. 2: To improve readability, the contributions could be displayed as a list.
- p. 2: $\pi$ is defined in a deterministic but used in a stochastic manner.

---

> ### Author Response · Authors · 2023-11-16
>
> > The relation between the hard constraint and the Gram-Schmidt process
>
> The constrained optimization problem in Eq. 1 shows our objective mathematically. However, we realize the hard constraint by applying the Gram-Schmidt process on the generated representations from the experts to orthogonalize them, hence satisfying the constraint. We modified the main paper adding two sentences clarifying this part. Thank you for pointing this out.
>
> > PPO horizontal line
>
> We included the single-task performance as a dashed line for two reasons. First, for the sake of clarity of the plots since we have many baselines. Second, it's common in the multi-task Learning literature to consider the single-task performance as a performance bound either as an upper bound in [1] or as a lower bound in [2], hence we care more about the final performance of the single-task agent.
>
> > The number of experts k
>
> For MiniGrid, the number of experts $k$ (generating the $k$ representations) is $2$, $3$, and $4$ for MT3, MT5, and MT7, respectively. On the other hand, the number of experts is $4$ and $6$ for MT10-rand and MT50-rand of the MetaWorld benchmark, respectively. We have clarified that point in the revised version of the paper.
>
> > How does the number of experts impact the scalabiltiy of the proposed approach?
>
> In Fig. 4, we showed the effect of increasing the number of experts regarding the performance. On average, we can clearly see that the performance is increasing much faster than the MOE baseline. However, this comes with a cost. The more experts we use, the higher the time complexity and the memory requirement. So, there is definitely a trade-off between the representation capacity and the time and memory requirements.
>
> > The computation and memory requirments
>
> The difference between MOORE and MOE is in the Gram-Schmidt stage where we orthogonalize the $k$-representations. The time complexity of the Gram-Schmidt process is $T = O(k^{2}\times d$) [3,4], where $d$ is the representation dimension and $k$ is the number of experts. MOORE and MOE can be considered as a soft-MOE because they both compute the whole $k$ representations from all the experts and then aggregate them. MOORE and MOE have the same memory requirement of storing the parameters of the mixture of experts. The memory requirement is also similar in PaCo; however, in the MetaWorld experiments, we exhibit better performance using fewer experts. This shows that diversity plays an important role in maximizing the representation capacity. In the revised version, we have provided a clear discussion about the computation and memory requirements in Appendix C.
>
> > To improve readability, the contributions could be displayed as a list.
>
>   Thank you for this comment. We have updated the contribution paragraph in the revised version by adding an index for each contribution.

---

> ### Author Response · Authors · 2023-11-16
>
> > Introduction modifications and motivation
>
> In this work, our goal is to learn a set of diverse representations that cover the common objects, skills, and properties shared across tasks. We want to remark that sharing knowledge is known to be key to the success of the MTRL algorithm, as discussed in multiple works in the MTRL literature [1,2,5,6].
>
> > Empirical results in the contribution list
>
> Thank you for your suggestion. We have added the empirical results regarding the state-of-the-art performance of MOORE on MetaWorld as part of the contribution list.
>
> > Enhancing Figure 1
>
> Thank you for your feedback. In the revised version, we have updated the visual illustration of MOORE in Figure 1 by demonstrating in detail the computation happening inside the output head.
>
> > MOORE and the RL algorithms
>
> We appreciate your comment. We stated that fact in Appendix A.1.2 and A.2.2. However, we have also clarified that point in Section 5 in the revised version of the paper.
>
> > Baseline
>
> All baselines are well-known in the MTRL literature. We provided a brief description of each baseline in the paper. Moreover, in the revised version, we have added the implementation details about MTPPO and MTSAC in Appendix A.1.2 and A.2.2 since they are a central part of our approach and the other related baselines.
>
> > Figure placement
>
> Thank you. We have enhanced the figure placement in the revised version of the paper.
>
> > Limitations should be stated
>
> MOORE is considered as a soft-MOE algorithm because it requires computing the whole $k$ representations from all experts and then aggregating them, compared to the sparse selection of one expert as done in the sparse-MOE algorithms. Thus, MOORE has the limitation of potentially suffering from high time complexity, resulting in a trade-off between the representation capacity and time complexity. In this work, our goal is to make use of the whole representational space spanned by all the diverse experts and consider them as a set of basis, letting the task-encoder produces the task-specific weights to aggregate those bases. As a future work, we can investigate the possibility of selecting a few orthogonal experts like in the sparse-MOE algorithms. In the revised version of the paper, we have stated this limitation in the conclusion.
>
> > Related works improvement
>
> Thank you for the additional reference. Considering the focus on MTRL in our work, we focus on reviewing related works on a mixture of experts in the domain of MTRL. Nevertheless, we have added additional references about the mixture of experts and diversity in RL in the revised version of the paper.
>
> > Open-sourcing the implementation
>
> We plan to open-source our implementation upon acceptance.
>
>
> > $\pi$ is defined in a deterministic but used in a stochastic manner.
>
> Yes, in the preliminaries, we used a deterministic form of the policy for simplicity. However, we agree with the Reviewer that presenting it in the stochastic form can enhance the clarity of the paper. In the revised version of the preliminaries, we have defined the policy in the stochastic setting.
>
> [1] Sodhani, Shagun, et al. "Multi-task reinforcement learning with context-based representations." ICML (2021).
>
> [2] D'Eramo, Carlo, et al. "Sharing knowledge in multi-task deep reinforcement learning." ICLR (2019).
>
> [3] Golub, Gene H., and Charles F. Van Loan. Matrix computations. JHU press, (2013).
>
> [4] Mashhadi, Peyman Sheikholharam, et al. "Parallel orthogonal deep neural network." Neural Networks (2021).
>
> [5] Yu, Tianhe, et al. "Gradient surgery for multi-task learning." NeurIPS (2020)
>
> [6] Hessel, Matteo, et al. "Multi-task deep reinforcement learning with popart." AAAI (2019).

---

> > ### Comment · Reviewer_stvx · 2023-11-23
> >
> > Thank you for the clarifications and providing a revised version.

---

### Official Review · Reviewer_pAYF · 2023-11-01

**Soundness:** 3 good
**Presentation:** 2 fair
**Contribution:** 2 fair
**Rating:** 5
**Confidence:** 5

**Summary:**

The paper aims to learn shared representations for multi-task reinforcement learning. The authors propose that these representations should be as diverse as possible. They define a contextual MDP called the Stiefel Contextual Markov Decision Process (SC-MDP) where the states are projected into the Stiefel manifold. The property of a Stiefel manifold is all elements can be represented as k-orthonormal representations i.e. representations that comprise k orthonormal vectors, each of dimension d. They impose a hard constraint on the optimization using the Gram-Schmidt process. They achieve competitive results on minigrid and metaworld.

**Strengths:**

(1) The use of the Gram-Schmidt process to enforce a hard constraint of orthonormality on the optimization is interesting.

(2) If the vectors are orthonormal, the state will be represented using k diverse vectors, providing a greater span of the state space. This bears fruit as empirically, they perform better when the vectors are orthonormal i.e. when the hard constraint is applied.

(3) They introduce the SC-MDP formulation where each state can be mapped to an element in the Stiefel manifold.

**Weaknesses:**

(1) Please provide an algorithm box. I am not sure why these different vectors are called experts. Is it for every step of the convex optimization, you perform the Gram-Schmidt orthonormalization like projected gradient descent? Or do you perform the normalization towards the end?

(2) Say you obtain $U^* = [u_1, u_2, ..., u_k]$ by minimizing $J(\Theta)$. Are these k vectors linearly independent? Gram-Schmidt process works on linearly independent vectors. Moreover, the vectors that you get after the process, $V^*= [v_1, v_2, ..., v_k]$ span the same space as $U^*$ but the $w_c$ for both will be different. I think these confusions can be cleared if you provide an algorithm box.

(3) You mention that MOORE even outperforms the single task performance is a few cases. But in those cases, most baselines also outperform the single task performance.

(4) Please address the questions in the following section.

**Questions:**

(1) What do you mean by frames in Definition 4.1?

(2) How do you parameterized the critic? You have only mentioned the parameterization of actor.

(3) Explain MT-PPO - How is PPO adapted for multi-task RL?

(4) What do you mean by single-head and multi-head architectures? Do you refer to single-head and multi-head attention?

(5) For transfering experts, you write "we transfer experts learnt on MT3 to MT5 (MT3 $\rightarrow$ MT5) and on MT5 to MT7 (MT5 $\rightarrow$ MT7) ". But in the next sentance you write "First, we train on base tasks (intersection of two scenarios), and then we transfer the learnt experts on novel tasks (the difference between the two scenarios)". Both statements sort of contradict each other.

---

> ### Author Response · Authors · 2023-11-16
>
> > What do you mean by frames in Definition 4.1?
>
> We use the term orthonormal $k$-frame, in the definition, to represent a set of orthonormal vectors. If the reviewer prefers, we can replace the term "frames" with "vectors".
>
> > How do you parameterized the critic? You have only mentioned the parameterization of actor.
>
> We parameterize the critic similar to the actor. Only in MetaWorld, instead of mapping the state only, we map the state-action by the mixture of experts, followed by the application of the Gram-Schmidt process to orthogonalize the representations. Then, the task-encoder generates the task-specific weights $w_{c}$ given the context $c$ (e.g., as task-id). The task-specific representation $v_{c}$ of the state-action can be computed using the task-specific weight $w_{c}$ and the set of representations $V_{s,a}$. The $Q$-value is then calculated either using a single-head architecture where we concatenate the task-id with the task-specific representation as an input to a single output module $f_{\theta}$, or using a multi-head architecture where the output module consists of multiple task-specific output modules where the corresponding output module is selected using the context $c$ (e.g., as task-id). In the revised version, we provide Alg. 1 and Alg. 2 of the MOORE forward function for the Actor and of the critic, respectively.
>
> > Explain MT-PPO - How is PPO adapted for multi-task RL?
>
> We adapt PPO to the MTRL setting by training the actor (policy) and critic (value function) by minimizing their corresponding loss functions averaged on transitions sampled from all tasks. We have added the PPO adaptation details in Appendix A.1.2.
>
> > Single-head and Multi-head
>
> The discussion of single-head and multi-head is highlighted in Appendix A.1.2. In the revised version, we enhanced the discussion in the Appendix and added a clear sentence in section 4.2 of the methodology. Here, we provide a clear distinction between the single-head and multi-head architectures. In multi-task, there are two common architectures: single-head, and multi-head. The difference between them is highly dependent on how we condition our network. In single-head architecture, the context $c$ (e.g. task-id) is usually concatenated with the input and fed to the network. Since MOORE has a task-agnostic representation block, we concatenate the context $c$ with the output of the representation block (aggregated representation) $v_{c}$. On the contrary, the multi-head consists of multiple task-specific output modules $f_{\theta} = [f_{\theta_{1}}, .., f_{\theta_{|\mathcal{C}|}}]$ where the context $c$ is responsible for selecting the corresponding task-specific output module $f_{\theta_{c}}$. This is also valid for the baselines.
>
> > Transfer learning study
>
> We agree that this part needs more elaboration. We have added Appendix B.2 in the revised version of the paper to elaborate more on the experimental details of the transfer learning study. In our transfer learning study, we assess the transfer capability of our approach in utilizing the diverse representations, learned on a set of *base* tasks, for a set of *novel* but related tasks. We evaluate our approach, MOORE, against the MOE baseline on MiniGrid. We refer to the transfer learning adaptation of our approach as **Transfer-MOORE**, and **Transfer-MOE** for the MOE baseline.
>
> We conducted two experiments based on the sets of tasks defined on MiniGrid (MT3, MT5, and MT7). We show the empirical results on two transfer learning scenarios where we transfer a set of experts learned on MT3 to MT5 (MT3 $\rightarrow$ MT5), and on MT5 to MT7 (MT5 $\rightarrow$ MT7). It is worth noting that MT3 is a subset of MT5, and MT5 is a subset of MT7. We consider the intersection between every two sets (MT3 and MT5 or MT5 and MT7) as base tasks while considering the difference as novel tasks. For instance, in the MT3$\rightarrow$MT5 scenario, the base tasks are LavaGap,  RedBlueDoors, and Memory (common for MT3 and MT5), while having DoorKey, and MultiRoom as novel tasks (only in MT5).
>
> Transfer-MOORE learns the base tasks and then transfers the learned mixture of experts for learning the novel ones. While learning the novel tasks, the mixture of experts is frozen but the task encoder and output head are being trained. On the contrary, MOORE is only trained on novel tasks from scratch. This also holds for MOE and Transfer-MOE. In this study, we are employing a multi-head architecture, the output head consists of completely decoupled task-specific output modules. Therefore, we train only the task encoder and the task-specific output modules for the novel tasks from scratch, keeping the mixture of experts frozen and the base task-specific heads too.

---

> ### Author Response · Authors · 2023-11-16
>
> > Algorithm box
>
> To clear any ambiguity, we provide now an algorithm box in the appendix of the revised version, regarding the adaptation of the actor (Algorithm 1) and critic (Algorithm 2) using MOORE. We consider the state encoders as experts. At each step, the generated representations $U$ from the experts (state encoders) are orthogonalized by applying the Gram-Schmidt process to satisfy the hard constraint in Eq. 1. We aggregate the orthogonal representations $V$ using the task-specific weights $w_{c}$.
>
> > Are these k vectors linearly independent? Gram-Schmidt process works on linearly independent vectors.
>
> We agree that the input vectors to the Gram-Schmidt process should be linearly independent. In this work, we cannot guarantee the independence of the vectors before the process; however, we have not encountered any instability during learning. We believe that the reason is that the Gram-Schmidt is a differentiable process and the gradients are actually contributing to learning the experts, hence favoring the independence of the input vectors.
>
> > You mention that MOORE even outperforms the single task performance is a few cases. But in those cases, most baselines also outperform the single task performance.
>
> Here, we highlighted that MOORE outperforms the single-task performance with a **significant margin**. The Reviewer correctly points out that also the other baselines manage to outperform the single-task performance in the same setting, although we can observe that they do it by a lower margin than ours. We agree that we should have stated this point clearer and we have done it in the revised version.

---

> > ### Comment · Reviewer_pAYF · 2023-11-20
> >
> > Thanks a lot for your clarifications. However, there are still some concerns.
> >
> > (1) You don't need to replace the term "frames" with "vectors" but be sure to explain the term you are using. In Linear Algebra, the term "vectors" is more common
> >
> > (2) In MTPPO, are you conditioning the policy and value function on the task id?
> >
> > (3) The transfer learning pipeline is clear now. But if MT3 is a subset of MT5, doesn't the intersection tasks simply mean MT3? If so, you don't need to even write the term intersection set as it only adds confusion.
> >
> > (4) I am still concerned about the use of Gram Schmidt process as there is no way to ensure that the input vectors $U_s$ are linearly independent. From the algorithm box, you are simply using a network to generate $U_s$ and applying the Gram-Schmidt process on top of it and hoping that by backpropagation, as the process is differentiable, it will train the network $h_\phi$ to generate linearly independent vectors. I am not fully convinced why this will work. Even from your comment, it looks like you are also not very sure if the input vectors are in fact linearly independent.
> >
> > (5) You say that MOORE outperforms the single-task performance by a significant margin. Agreed. But the baselines also outperform by a good margin. The curves look very close to each other. So how much is the margin? The only reason I am concerned about this is that mathematically, as the input might not be linearly independent, the produces experts are not orthogonal and so want to see if empirically, the proposed solution provides significant improvement.

---

> > > ### Author Response · Authors · 2023-11-21
> > >
> > > Thank you for your feedback.
> > >
> > > > (1) You don't need to replace the term "frames" with "vectors" but be sure to explain the term you are using. In Linear Algebra, the term "vectors" is more common
> > >
> > > To remove any confusion, we decided to replace the term "frames" with "vectors" in Definition 4.1.
> > >
> > > > (2) In MTPPO, are you conditioning the policy and value function on the task id?
> > >
> > > In MTTPO, we condition both the policy and the value function on the task ID. Both of them can utilize one of two architectures, single-head or multi-head. For single-head architecture, we concatenate the task-id with the representation generated from the task-agnostic representation block and then we feed it to the single output module $f_{\theta}$. On the other hand, in the multi-head architecture, we instead use the task-id to select the task-specific output module $f_{\theta_{c}}$, then we feed the representation generated from the task-agnostic representation block to the selected output module. Since we build all the other baselines, including MOORE, on top of MTTPO, we follow the same procedure. The only difference is that the task-agnostic representation block consists of a mixture of experts, hence the input to the output module is the aggregated representation (task-specific representation) $v_c$ from all experts. For the sake of completeness, the task-id is represented by a one-hot vector.
> > >
> > > > (3) The transfer learning pipeline is clear now. But if MT3 is a subset of MT5, doesn't the intersection tasks simply mean MT3? If so, you don't need to even write the term intersection set as it only adds confusion.
> > >
> > > Yes, the intersection between the MT3 and MT5 is the MT3 set. That's why we annotate the transfer scenario as MT3$\rightarrow$MT5. Since we learn the base tasks (represented by the MT3 task set) and then we train on the difference between the sets as novel tasks. The final transfer model is able to perform inference on all tasks in MT5 since the mixture of experts is frozen during the transfer process, and the output modules are decoupled as we employ a multi-head architecture. It is worth noting, as stated in Appendix B.2, that the task-encoder is a single linear layer without a bias term. Therefore, we can extend it easily without affecting the encoding of the base tasks.
> > > Nevertheless, we agree that the intersection set can be confusing and we decided to remove it in the revised version.
> > >
> > > > (4) I am still concerned about the use of Gram Schmidt process as there is no way to ensure that the input vectors $U_s$ are linearly independent. From the algorithm box, you are simply using a network to generate $U_s$ and applying the Gram-Schmidt process on top of it and hoping that by backpropagation, as the process is differentiable, it will train the network $h_{\phi}$ to generate linearly independent vectors. I am not fully convinced why this will work. Even from your comment, it looks like you are also not very sure if the input vectors are in fact linearly independent.
> > >
> > > It is true that the Gram-Schmidt process requires a linear independence of its input vectors. In our experiments, we have evaluated our approach on $6$ scenarios on MiniGrid with $30$ runs each, and On MetaWorld, on $2$ settings (MT10-rand, MT50-rand) with $10$ different runs each. Across all our experimental campaigns, we have never encountered a single instability using the Gram-Schmidt, which indicates that the generated representations $U_s$ have never happened to be linearly dependent. In fact, we have faced only a single case of instability which happened when we initialized the experts with the same set of parameters. Based on this observation, we have decided to randomly initialize the parameters of the experts for all experiments.
> > >
> > > Even if instability due to Gram-Schmidt has not been a problem for us, we want to clarify that there are some workarounds to prevent the detrimental effect of potentially linearly dependent vectors. For instance, one method is to perturb the vector(s) that cause the dependency issue by a small constant $\epsilon$.
> > >
> > > Finally, we want to point out that the Gram-Schmidt process is the way we choose to realize the hard constraint in Eq. 1, and that works well for us. Future works can investigate alternative approaches to learn a set of orthogonal experts that capture diverse representations of the common properties shared across tasks.

---

> > > ### Author Response · Authors · 2023-11-21
> > >
> > > > (5) You say that MOORE outperforms the single-task performance by a significant margin. Agreed. But the baselines also outperform by a good margin. The curves look very close to each other. So how much is the margin? The only reason I am concerned about this is that mathematically, as the input might not be linearly independent, the produces experts are not orthogonal and so want to see if empirically, the proposed solution provides significant improvement.
> > >
> > > For clarification, we provide performance tables of all algorithms for the discussed MiniGrid experiments. We indicate the mean and the confidence interval of the average return metric every $10$ epochs. Moreover, we provide the margin (delta) to the single-task performance of each algorithm. As shown, MOORE outperforms all the baselines not just in terms of the final performance but also in terms of sample efficiency. In terms of the margin to the single-task performance, we consistently have a larger margin than the other baselines, especially in MT5.
> > >
> > > *We thank the Reviewer for the valuable feedback that we believe helped to improve our paper. Therefore, we kindly ask the Reviewer to consider providing a reassessment of our work in light of these improvements.*
> > >
> > > ---
> > > **MT3**
> > >
> > > | Algorithms | 10 | 20 | 30 | 40 | 50 | 60 | 70 | 80 | 90 | 100 | Margin |
> > > | -------- | -------- | -------- | -------- | -------- | -------- | -------- | -------- | -------- | -------- | -------- | -------- |
> > > | MTPPO | 0.17 $\pm$ 0.02 | 0.22 $\pm$ 0.03 | 0.31 $\pm$ 0.04 | 0.36 $\pm$ 0.04 | 0.43 $\pm$ 0.05 | 0.48 $\pm$ 0.06 | 0.51 $\pm$ 0.05 | 0.57 $\pm$ 0.05 | 0.59 $\pm$ 0.05 | 0.62 $\pm$ 0.05 | 0.04 |
> > > | PCGrad | 0.17 $\pm$ 0.02 | 0.27 $\pm$ 0.04 | 0.36 $\pm$ 0.05 | 0.46 $\pm$ 0.05 | 0.52 $\pm$ 0.05 | 0.57 $\pm$ 0.05 | 0.58 $\pm$ 0.05 | 0.61 $\pm$ 0.05 | 0.66 $\pm$ 0.05 | 0.7 $\pm$ 0.04 | 0.12 |
> > > | MOE | 0.17 $\pm$ 0.02 | 0.27 $\pm$ 0.04 | 0.37 $\pm$ 0.05 | 0.45 $\pm$ 0.05  | 0.5 $\pm$ 0.05 | 0.54 $\pm$ 0.05 | 0.58 $\pm$ 0.05  | 0.61 $\pm$ 0.06  | 0.63 $\pm$ 0.05 | 0.65 $\pm$ 0.05 | 0.07 |
> > > | MOORE (ours) | **0.23 $\pm$ 0.02** | **0.31 $\pm$ 0.05** | **0.43 $\pm$ 0.05** | **0.51 $\pm$ 0.04**  | **0.6 $\pm$ 0.04**  | **0.64 $\pm$ 0.05**  | **0.67 $\pm$ 0.03** | **0.7 $\pm$ 0.04** | **0.71 $\pm$ 0.03** | **0.73 $\pm$ 0.03** | **0.15** |
> > >
> > > ---
> > > **MT5**
> > >
> > > | Algorithms | 10 | 20 | 30 | 40 | 50 | 60 | 70 | 80 | 90 | 100 | Margin |
> > > | -------- | -------- | -------- | -------- | -------- | -------- | -------- | -------- | -------- | -------- | -------- | -------- |
> > > | MTPPO | 0.14 $\pm$ 0.02 | 0.24 $\pm$ 0.03 | 0.33 $\pm$ 0.04 | 0.39 $\pm$ 0.04 | 0.46 $\pm$ 0.04 | 0.52 $\pm$ 0.04 | 0.55 $\pm$ 0.04 | 0.57 $\pm$ 0.04 | 0.58 $\pm$ 0.04 | 0.6 $\pm$ 0.04 | -0.04 |
> > > | PCGrad | 0.22 $\pm$ 0.03 | 0.35 $\pm$ 0.03 | 0.45 $\pm$ 0.04 | 0.5 $\pm$ 0.04 | 0.56 $\pm$ 0.05 | 0.59 $\pm$ 0.05 | 0.62 $\pm$ 0.05 | 0.66 $\pm$ 0.05 | 0.69 $\pm$ 0.04 | 0.68 $\pm$ 0.05 | 0.04 |
> > > | MOE | 0.19 $\pm$ 0.03 | 0.32 $\pm$ 0.04 | 0.43 $\pm$ 0.04 | 0.47 $\pm$ 0.05  | 0.53 $\pm$ 0.04 | 0.57 $\pm$ 0.05 | 0.6 $\pm$ 0.05  | 0.64 $\pm$ 0.05  | 0.65 $\pm$ 0.06 | 0.66 $\pm$ 0.05 | 0.02 |
> > > | MOORE (ours) | **0.26 $\pm$ 0.03** | **0.42 $\pm$ 0.04** | **0.54 $\pm$ 0.05** | **0.61 $\pm$ 0.05**  | **0.67 $\pm$ 0.05**  | **0.69 $\pm$ 0.05**  | **0.71 $\pm$ 0.04** | **0.73 $\pm$ 0.04** | **0.75 $\pm$ 0.05** | **0.76 $\pm$ 0.04** | **0.12** |
> > >
> > > ---
> > > **MT7**
> > >
> > > | Algorithms | 10 | 20 | 30 | 40 | 50 | 60 | 70 | 80 | 90 | 100 | Margin |
> > > | -------- | -------- | -------- | -------- | -------- | -------- | -------- | -------- | -------- | -------- | -------- | -------- |
> > > | MTPPO | 0.12 $\pm$ 0.02 | 0.25 $\pm$ 0.03 | 0.34 $\pm$ 0.04 | 0.41 $\pm$ 0.04 | 0.47 $\pm$ 0.04 | 0.5 $\pm$ 0.04 | 0.53 $\pm$ 0.04 | 0.56 $\pm$ 0.05 | 0.6 $\pm$ 0.04 | 0.61 $\pm$ 0.04 | -0.03 |
> > > | PCGrad | 0.18 $\pm$ 0.02 | 0.35 $\pm$ 0.03 | 0.46 $\pm$ 0.03 | 0.53 $\pm$ 0.04 | 0.57 $\pm$ 0.04 | 0.62 $\pm$ 0.04 | 0.64 $\pm$ 0.04 | 0.67 $\pm$ 0.04 | 0.69 $\pm$ 0.04 | 0.71 $\pm$ 0.04 | 0.07 |
> > > | MOE | 0.18 $\pm$ 0.02 | 0.32 $\pm$ 0.03 | 0.41 $\pm$ 0.05 | 0.49 $\pm$ 0.04  | 0.53 $\pm$ 0.05 | 0.59 $\pm$ 0.04 | 0.62 $\pm$ 0.05  | 0.65 $\pm$ 0.04  | 0.67 $\pm$ 0.04 | 0.69 $\pm$ 0.03 | 0.05 |
> > > | MOORE (ours) | **0.27 $\pm$ 0.02** | **0.43 $\pm$ 0.03** | **0.55 $\pm$ 0.03** | **0.61 $\pm$ 0.03**  | **0.65 $\pm$ 0.04**  | **0.68 $\pm$ 0.03**  | **0.71 $\pm$ 0.04** | **0.74 $\pm$ 0.04** | **0.74 $\pm$ 0.03** | **0.75 $\pm$ 0.03** | **0.11** |

---

> ### Comment · Reviewer_pAYF · 2023-11-22
>
> Thanks a lot for the additional clarifications and results. I see that the improvements of MOORE over PCGrad are small. Moreover, if the Gram-Schmidt process does not generate orthogonal vectors as their inputs are linearly dependent, it might not always lead to instability in training, it will just lead to poor outputs. Empirically, the method might overperform the baselines, still, if applied correctly, i.e. if the vectors are indeed orthogonal, the results can be better.
>
> I believe that the idea of using Gram Schmidt process is interesting and novel and has some empirical improvements over the baselines so I have increased my score.

---

> > ### Author Response · Authors · 2023-11-23
> >
> > We thank you for the fruitful discussion and the reconsideration of your assessment.
> >
> > Motivated by this discussion, we have conducted further investigation to address the Reviewer's concern about the use of the Gram-Schmidt process. We have run all three scenarios, namely MT3, MT5, and MT7, again, while checking the minimum angle between every two experts. During the training, we have observed that it is always the case that the representations are orthogonal.
> >
> > In addition, in our implementation, we always normalize the vectors after projection. If two or more vectors happen to be linearly dependent, the projection results in the zero vector. Accordingly, the normalization process would produce Nan values, which we have not encountered in our experiments, except in the single case where we have initialized different experts with equal parameters. Nevertheless, we understand the Reviewer's concern about the possibility of incurring linearly dependent vectors. In such a case, we remark that some workarounds could be applied, like perturbing the dependent vectors by a small coefficient $\epsilon$, as mentioned before.

---

### Official Review · Reviewer_gg3H · 2023-11-01

**Soundness:** 3 good
**Presentation:** 3 good
**Contribution:** 2 fair
**Rating:** 5
**Confidence:** 4

**Summary:**

The authors introduce a novel representation learning method for multi-task reinforcement learning (MTRL) that promotes orthogonal representations to enhance diversity. Initially, they present a Stiefel Contextual Markov Decision Process (SC-MDP) to interpret the orthogonal representation spaces. Following this, the authors propose a constrained optimization problem that ensures orthogonality in the representations and employ the Gram-Schmidt process to address the constraints. Empirical results demonstrate that their proposed method, MOORE, consistently surpasses previous MTRL algorithms across two MTRL benchmarks.

**Strengths:**

The paper proposes an interesting approach to promote the diversity of representation with Gram-Schmidt process and shows empirical improvements on two MTRL benchmarks.

**Weaknesses:**

The paper doesn't introduce a method that is fundamentally distinct from the approach in prior work[1], which utilizes a representative parameter set {$\phi_1,...,  \phi_k$} and task-specific weight $w_c$ derived from the task-id. The proposed structure for the policy appears to be merely different combinations of the existing learnable parameters discussed in paper[1]. While Paco[1] employs a linear combination of the representative parameters with the task-specific weight as the policy parameters, this paper adopts the same strategy to compute an additional input for policy inputs combined with the state. Moreover, although the authors argue that orthogonality results in more diverse representations, there's an absence of theoretical backing or ablation studies to corroborate this claim. Given that the vector $k$-representations $U_s$ is contingent upon the state $s$, the authors cannot guarantee representational diversity at the task level, even if state-level orthogonality is ensured. Additionally, the paper does not address issues related to complexity and memory requirements. Storing a set of parameters for representation implies that the proposed "MOORE" method could demand significant memory and computational resources. Finally, the authors should benchmark their method against the MT50 benchmark from Meta-World, as all their experiments are conducted on a limited task set size.

[1] Sun, Lingfeng, et al. "PaCo: Parameter-Compositional Multi-Task Reinforcement Learning." Advances in Neural Information Processing Systems 35 (2022): 21495-21507.

**Questions:**

1. The Gram-Schmidt process can yield different vectors based on which vector is chosen as the first. How do the authors determine the initial vector for the Gram-Schmidt process?

2. Can you provide the results for the MT50 benchmark?

3. How many representations $k$ are used in each experiment?

4. Could you elucidate the distinction between MOORE and Transfer-MOORE?

---

> ### Author Response · Authors · 2023-11-16
>
> > How do the authors determine the initial vector for the Gram-Schmidt process?
>
> - Thank you for raising this point. In MOORE, we consider always the first expert's representation as the initial vector for the Gram-Schmidt process. We agree that the process will yield different set of orthonormal vectors depending on the initial selected vector. We argue that it does not matter in our case since the representations are actually generated from the mixture of experts which are being trained.
> - In the Appendix D of the revised version, we have added an ablation study on MT5 of MiniGrid where we utilize $3$ experts. We provided the variations of MOORE based on the initial vector selected for the Gram-Schmidt process. The study shows that the performance is almost the same while varying the selected initial vector, supporting our claim that the initial vector does not matter since the experts are being trained.
>
> > Can you provide the results for the MT50 benchmark?
>
> We agree about the importance of testing MOORE on MT50, and we apologize for not having done it in the original submission due to lack of time. We have added the comparison between MOORE and the MOE baseline on MT50 for 10 epochs in the experimental section of the revised version. The experiments are still running to obtain results for 20 epochs, which we plan to publish in a new revision in the upcoming days, in order to be able to compare with the already available results of the other baselines in [1]. Notably, as shown in the new Figure 6b, the performance of MOORE obtained after 10 epochs is already superior than the state-of-the-art of PaCo [1], i.e., $57.3%$. Moreover, our performance has been achieved using only $6$ experts in comparison to the state-of-the-art PaCo which uses $20$ experts.
>
> >How many representations k are used in each experiment?
>
> For MiniGrid, the number of experts $k$ (generating the $k$ representations) is $2$, $3$, and $4$ for MT3, MT5, and MT7, respectively. On the other hand, the number of experts is $4$ and $6$ for MT10-rand and MT50-rand of the MetaWorld benchmark, respectively. We have clarified that point in the revised version of the paper.
>
> >Could you elucidate the distinction between MOORE and Transfer-MOORE?
>
> In our transfer learning study, we assess the transfer capability of our approach in utilizing the diverse representations, learned on a set of *base* tasks, for a set of *novel* but related tasks. We evaluate our approach, MOORE, against the MOE baseline on MiniGrid. We refer to the transfer learning adaptation of our approach as **Transfer-MOORE**, and **Transfer-MOE** for the MOE baseline. Transfer-MOORE learns the base tasks and then transfer the learned mixture of experts for learning the novel ones. While learning the novel tasks, the mixture of experts is frozen but the task encoder and output head are being trained. On the contrary, in this study, MOORE is only trained on the novel tasks from scratch. This also holds for MOE and Transfer-MOE. We also added a subsection in the Appendix A.4 elaborating more on the experimental details of the transfer learning study.
>
> >Similarity to Paco.
>
> PaCo [1] interpolates a task-specific policy from a subspace of policies spanned by a set of parametric policies while utilizing task-specific weights $w_{c}$. Hence, PaCo searches for the task-specific policy in the parametric space, not in the feature/representation space. On the other hand, MOORE utilizes a set of experts (state-encoders for actor or state-action-encoders for critic) to generate a set of orthogonal representations that spans a representation subpsace where each task can interpolate the relevant representation. Using the task-id is not crucial in MOORE, nor in PaCo. It is just an example on how we can interpolate task-specific information. For instance, CARE [2] uses the task description instead of the task-id while doing attention mechanism to produce set of representations. Moreover, PaCo applies some tricks (loss-maskout and w-reset) to stabilize the MTRL training process, however, our approach does not employ any tricks.
>
> >Theoretical backing or ablation studies to corroborate this claim. Given that the vector
> $k$-representations $U_{s}$ is contingent upon the state $s$, the authors cannot guarantee representational diversity at the task-level, even if state-level orthogonality is ensured.
>
> Thank you for raising this interesting point. We want to clarify that we are focusing on representing each state by a set of diverse representations, and we agree that we have not targeted the diversity at the task-level. Nevertheless, as shown in the experiemental section, the orthogonality condition succeeds to make the representations at the state-level diverse and outperforming the other baselines. In addition, in Fig. 4, we can deduce that the diversity manages to increase the state representation capacity as we increase the number of experts in comparison to not enforcing diversity.

---

> ### Author Response · Authors · 2023-11-16
>
> >Complexity and memory requirements
>
> - The difference between MOORE and MOE is in the Gram-Schmidt stage where we orthogonalize the k-representations. The time complexity of the Gram-Schmidt process is $T = O(k^{2}\times d$) [3,4], where $d$ is the representation dimension and $k$ is the number of experts.
> - MOORE and MOE can be considered as a soft-MOE because they both compute the whole $k$ representations from all the experts and then aggregate them.
> - MOORE and MOE have the same memory requirement of storing the parameters of the mixture of experts. The memory requirement is also similar in PaCo; however, in the MetaWorld experiments, we exhibit better performance using fewer experts. This shows that diversity plays an important role in maximizing the representation capacity. In the revised version, we have provided a detailed discussion about the computation and memory requirements in Appendix C.
>
> [1] Sun, Lingfeng, et al. "PaCo: Parameter-Compositional Multi-Task Reinforcement Learning." NeurIPS (2022).
>
> [2] Sodhani, Shagun, Amy Zhang, and Joelle Pineau. "Multi-task reinforcement learning with context-based representations." ICML (2021).
>
> [3] Golub, Gene H., and Charles F. Van Loan. Matrix computations. JHU press, (2013).
>
> [4] Mashhadi, Peyman Sheikholharam, et al. "Parallel orthogonal deep neural network." Neural Networks (2021).

---

### Author Response · Authors · 2023-11-16
**Comments on the revised paper**

We thank the Reviewers for their valuable feedback. Our revised version incorporates significant changes aimed at addressing the raised concerns. We believe that the current version of our work is a substantial improvement over the initial submission. We kindly request all Reviewers to take this into consideration. To enhance clarity, we have highlighted the modifications in blue. In summary, the revised paper includes:
- Enhanced the visual illustration of MOORE in Fig. 1.
- Clarified the need of the Gram-Schmidt process for satisfying the hard constraint in Eq. 1.
- Clarified the transfer learning ablation study in Sec. 5.1.1 and Appendix B.2.
- Stated the number of experts for all experiments.
- Added the MT50 experiment in Fig. 6(b).
- Discussed the limitations in the conclusion (Sec. 6).
- Highlighted the adaptation of PPO and SAC to the MTRL setting in Appendix A.1.2 and A.2.2.
- Discussed the computation and memory requirements in Appendix C.
- Added an ablation study, in Appendix D, on the effect of the initial expert selected in the Gram-Schmidt process.

---

### Author Response · Authors · 2023-11-23
**MT50 experiments added**

We want to inform the Reviewers that, as promised, we have added the complete experimental results in MT50 in the revised version (Table 2). Notably, MOORE outperforms the best baseline PaCo in terms of success rate of approximately more than $15\\%$, confirming its good performance in challenging tasks and evincing its scalability to a large number of tasks. We thank the Reviewers for their feedback so far, which we believe helped enhance the quality of the paper.

---

### Meta-Review · Area_Chair_8zY4 · 2023-12-05

**Metareview:**

Summary: The authors propose a novel approach to multi-task reinforcement learning through shared representations by utilizing a Gram-Schmidt process to construct a set of orthogonal representations to ensure diversity. They evaluate on MiniGrid and MetaWorld and achieve SOTA performance on MetaWorld.

Strengths: I appreciate the presentation of the Stiefel Contextual Markov Decision Process (SC-MDP) as justification for requiring orthonormality of the learned vectors. SOTA results on MetaWorld plus the additional MT50 results are compelling, with comparison to competitive baselines.

Weaknesses: Baselines are somewhat lacking -- authors should compare to diversity-based methods like DIAYN and the algorithmic novelty in comparison to PaCo should be made more clear in the main paper. General clarity of the method and problem setting/framework could be improved.

What's missing: I would encourage authors to include a discussion and analysis of the true linear independence/measure the orthonomality of the learned vectors over training and for the final evaluated model. Also an ablation over number of experts and a computational cost comparison with baselines would be appreciated.

**Justification For Why Not Higher Score:**

While the reported results are significant, certain relevant baselines and comparisons are missing, and additional analysis of their proposed method could improve the paper.

**Justification For Why Not Lower Score:**

I believe the reported results, SC-MDP presentation and Gram-Schmidt process for producing orthonormal vectors were compelling enough for acceptance.

---

### Decision · Program_Chairs · 2024-01-16

Accept (poster)